# When Precision Meets Position:
# BFloat16 Breaks Down RoPE in Long-Context Training

**Haonan Wang**[*]                                          *haonan.wang@u.nus.edu*
*National University of Singapore*

**Qian Liu**                                                *liuqian@sea.com*
*Sea AI Lab, Singapore*

**Chao Du**                                                 *duchao@sea.com*
*Sea AI Lab, Singapore*

**Tongyao Zhu**                                             *tongyao.zhu@u.nus.edu*
*National University of Singapore*

**Cunxiao Du**                                              *ducx@sea.com*
*Sea AI Lab, Singapore*

**Kenji Kawaguchi**                                         *kenji@comp.nus.edu.sg*
*National University of Singapore*

**Tianyu Pang**[†]                                          *tianyupang@sea.com*
*Sea AI Lab, Singapore*

**Reviewed on OpenReview:** *https://openreview.net/forum?id=gwXfZ3xkUq*

## Abstract

Extending context window sizes allows large language models (LLMs) to process longer sequences and handle more complex tasks. Rotary Positional Embedding (RoPE) has become the de facto standard due to its relative positional encoding properties that benefit long-context training. However, we observe that **using RoPE with BFloat16 format results in numerical issues**, causing it to deviate from its intended relative positional encoding, especially in long-context scenarios. This issue arises from BFloat16's limited precision and accumulates as context length increases, with *the first token contributing significantly* to this problem. Despite its limitations, BFloat16 remains desirable for its computational efficiency, particularly given the substantial memory overhead required to extend the context window. To improve long-context training under BFloat16, we develop **AnchorAttention**, a plug-and-play attention method that enhances long-context capabilities, and speeds up training. AnchorAttention reduces unnecessary attention computations, maintains semantic coherence, and boosts computational efficiency by treating the first token as a shared anchor with a consistent position ID, making it visible to all documents within the training context. Experiments on three types of LLMs demonstrate that AnchorAttention significantly improves long-context performance and reduces training time by over 50% compared to standard full attention mechanisms, while preserving the original LLM's capabilities on general tasks.[1]

---

[*]Work done during Haonan Wang's internship at Sea AI Lab. [†]Correspondence to Tianyu Pang.
[1]*AnchorContext:* The implementation of AnchorAttention supports several popular models, using the FlashAttention2 and FlexAttention, and is available at https://github.com/haonan3/AnchorContext.

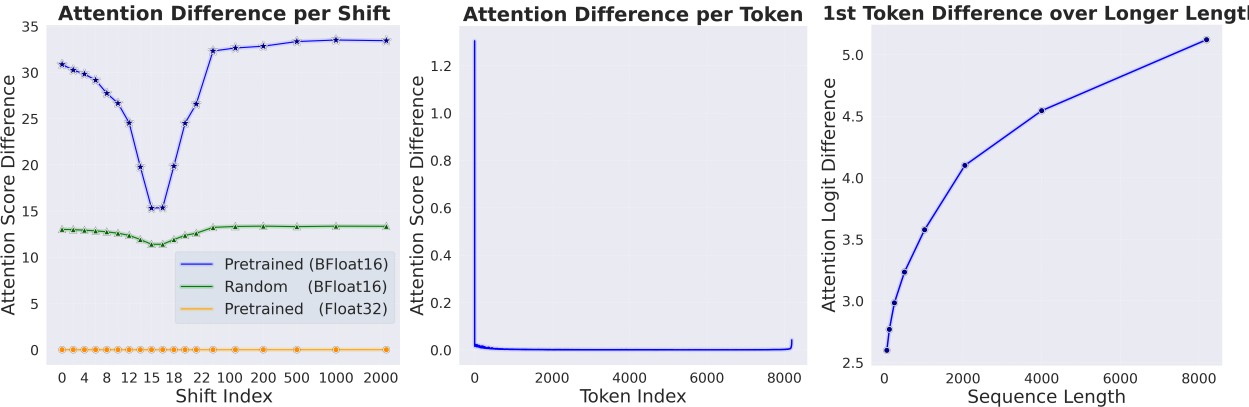

Figure 1: Effects of positional shifts on attention computations under different settings. **Left**: Attention difference $D$ (Eq. 4) plotted against varying positional shift $\Delta_1$ (with $\Delta_2 = 16$ fixed). Pretrained models under BFloat16 (blue line) exhibit significant discrepancies compared to Float32 (yellow line) and random initialization (green line), indicating that the relative positional encoding property of RoPE is broken under BFloat16 and that pretraining amplifies this effect. **Middle**: Per-token attention differences between $\Delta_1 = 0$ and $\Delta_2 = 16$, highlighting the first token accounts for most of the attention difference observed. **Right**: Attention logit difference (Eq. 5) for the first token as sequence length increases, showing increased discrepancies with longer sequences.

# 1 Introduction

In recent years, natural language processing has seen a surge in models that handle increasingly long sequence (Yang et al., 2024; Dubey et al., 2024; Jiang et al., 2023; Team et al., 2024). A 128K token context window allows large language models (LLMs) to handle complex tasks such as multi-document question answering (Wang et al., 2024a), repository-level code comprehension (Jimenez et al., 2024), and many-shot learning by capturing long-range dependencies (Agarwal et al., 2024), leading to more coherent and contextually relevant outputs (Mazumder & Liu, 2022).

In order to achieve long-context capabilities in LLMs, Rotary Position Embedding (RoPE) (Su et al., 2021) have emerged as the dominant backbone for positional encoding (Liu et al., 2023b). The success of RoPE is often attributed to its trigonometric (rotational) properties and its relative positional encoding, which enable models to avoid out-of-distribution (OOD) rotation angles (Wang et al., 2024b; Chen et al., 2023a; Peng et al., 2023; LocalLLaMA, 2023; Men et al., 2024). Theoretically, by adjusting the rotary frequencies, extended context position IDs (OOD) can be mapped to in-distribution ranges that have been sufficiently trained. In addition, due to its relative positional encoding properties, interpolating into the original familiar range allows the model to recognize the relative positions of input tokens in the extended context. As a result, a relatively small amount of additional training over long contexts enables LLMs to adapt to extended context lengths (Zhao et al., 2024a; Fu et al., 2024; Zhang, 2023). However, minimal long-context training remains challenging due to the quadratic increase in GPU memory consumption with context length (Xiong et al., 2023). To address this issue, Brain Floating Point (BFloat16)(Wang & Kanwar, 2019), commonly used during pre-training, is also adopted in the long-context training phase. Its use reduces memory bandwidth requirements without significantly impacting model accuracy(Kalamkar et al., 2019), making it ideal for managing the computational demands of long-context models.

Despite the computational advantages of BFloat16, we have identified a critical issue: **when combined with BFloat16, the relative positional encoding properties of RoPE are broken, especially in long-context scenarios.** As shown in Figure 1, this breakdown occurs because of BFloat16's limited precision. As the training window size increases, numerical errors accumulate, exacerbating the issue and resulting in a more substantial discrepancy. In contrast, this degradation disappears when using Float32, which maintains the integrity of RoPE's relative positional encoding. Our empirical observations confirm that this breakdown diminishes the benefits RoPE offers for long-context training.

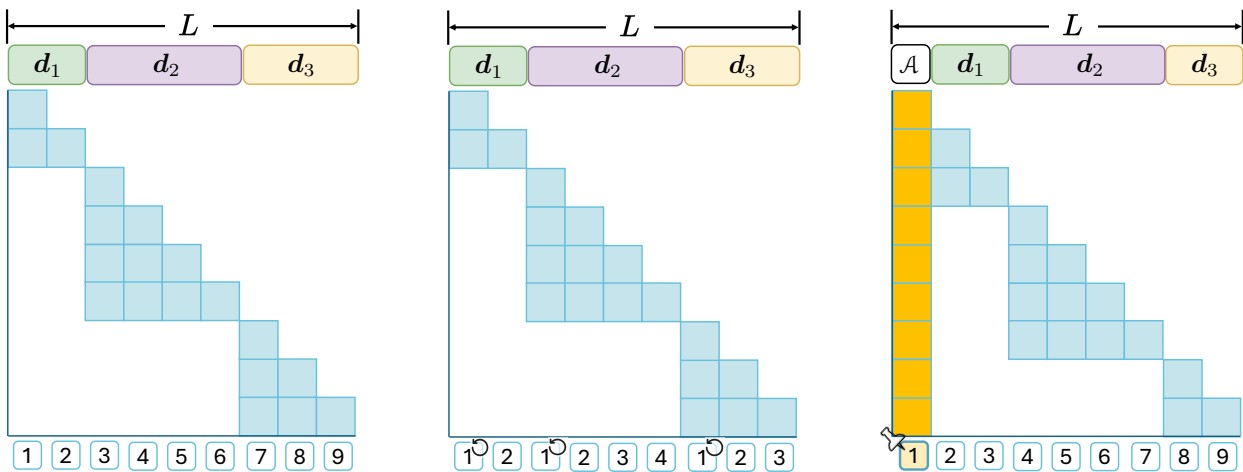

Figure 2: Illustrations of different attention paradigms. Left: Standard intra-document attention. Middle: Our improved version, intra-document attention with *position ID reset* per document. Right: **AnchorAttention** incorporating a shared anchor token, $\mathcal{A}$.

To improve long-context training, we investigated the source of the breakdown in RoPE's relative positional encoding and observed that the first token in the attention window contributes most significantly to this deviation. However, current methods do not explicitly consider this issue in their design. In the full attention mechanism, each token attends to all previous ones, resulting in cumulative deviations as the window size grows. Standard intra-document attention (Figure 2, left) uses cross-document masking, effectively converting the large window into multiple smaller ones. This approach introduces multiple first tokens for each small window, and assigning different position IDs to these first tokens causes inconsistencies in the model's positional understanding. Empirically, we found that simply resetting the position IDs to maintain consistency across windows (our version of intra-document attention, Figure 2, middle) improves long-context performance. This confirms that position ID inconsistencies are a key issue. Yet, resetting position IDs introduces a new problem: the model can only learn the full spectrum of rotational angles when processing data sequences that reach or exceed the maximum context length. Building on these insights, we propose **AnchorAttention**—a versatile and plug-and-play attention method that enhances long-context capabilities while accelerating the training process. As illustrated in Figure 2 (Right), the core innovation of AnchorAttention lies in treating the first token as a shared anchor: we always assign it the first position ID, making it visible to all documents within the context window while ensuring that tokens from different documents are invisible to each other. By having all documents share the same initial token, AnchorAttention eliminates inconsistencies while allowing the model to learn the full rotational span from sequences shorter than the context length. Additionally, AnchorAttention reduces the number of tokens involved in attention computations by not attending to all previous tokens, which helps prevent the rolling accumulation of numerical errors. Experimental results demonstrate that training with AnchorAttention consistently outperforms full attention, standard intra-document attention, and our improved intra-document attention on the long-context benchmark RULER, across lengths from 8K to 128K. On real-world long-context benchmarks like LongBench, AnchorAttention improves in-context learning performance while largely preserving the model's capabilities on general tasks such as MMLU and HellaSwag.

In summary, the main contributions of this paper are as follows:

- We found that the relative properties of RoPE are compromised under BFloat16 precision.
- We identified that the first token of a sequence contributes to the deviation of RoPE's relative properties, which should be preserved in theory. Moreover, this deviation becomes more pronounced with larger training window sizes.
- Based on these observations, we introduce a practical approach, **AnchorAttention**, for long-context continuous training, which improves the model's ability to handle long contexts, utilizes less than 50% of

the training time required by standard attention training, and requires minimal modifications to existing training pipelines.

## 2 Discrepancies in RoPE's Relative Positional Encoding

### 2.1 Background of Rotary Position Embedding (RoPE)

Modern LLMs are primarily based on the Transformer (Vaswani et al., 2017) architecture. One of its core components is the attention mechanism, which can be written as:

$$A_{ij} = \mathbf{q}_i^\top \mathbf{k}_j \tag{1}$$

$$\text{ATTN}(\mathbf{X}) = \text{softmax}(\mathbf{A} + \mathbf{M}^{-\infty}), \tag{2}$$

where the attention logit matrix $\mathbf{A} \triangleq (A_{ij}) \in R^{T \times T}$, the query and key vectors take the form $\mathbf{q}_i = \mathbf{W}_Q \mathbf{x}_i$ and $\mathbf{k}_j = \mathbf{W}_K \mathbf{x}_j$, respectively. $\mathbf{W}_Q, \mathbf{W}_K \in \mathbb{R}^{d \times d}$ are the query and key matrices, and $\mathbf{x}_i$, $\mathbf{x}_j$ represent the $i$-th and $j$-th token of $\mathbf{X} \in R^{T \times d}$. In the attention score matrix, $\text{ATTN}(\mathbf{X})$, we ignore the scaling factor $1/\sqrt{d}$ introduced by Vaswani et al. (2017) to simplify notation. The causal mask, $\mathbf{M}^{-\infty} \in \mathbb{R}^{T \times T}$, where $\mathbf{M}_{ij}^{-\infty} = -\infty$ if $i < j$ and $\mathbf{M}_{ij}^{-\infty} = 0$ otherwise, prevents each token from attending to future tokens by setting the upper triangular part of $\mathbf{A}$ to negative infinity.

Transformer architectures employ positional embeddings to encode sequence order information, which is essential for processing sequential data. RoPE (Su et al., 2021) are the currently most widely adopted encodings, especially in LLMs. RoPE acts on the query and key by splitting them into 2-dimensional chunks and rotating each chunk at a different frequency, which applies the rotation matrix into the calculation of the attention logit in Eq. 1, which can be written as:

$$A_{ij} = \underbrace{(R_{i,\theta} q_i)}^\top \underbrace{R_{j,\theta} k_j}_{\text{Applied before FlashAttention2, } \textbf{Float32}} = q_i^\top R_{\underline{j-i},\theta} k_j = \overbrace{q_i^\top R_{\underline{m},\theta} k_j}^{\text{Computed within FlashAttention2, } \textbf{BFloat16}},$$

where $m = j - i$ is the *relative distance* of $i$ and $j$. We leave the details of rotation matrix $R$ in Appendix A. RoPE achieves efficient relative position encoding by implementing it as absolute position encoding. This is done by applying a rotation matrix directly to the key and query vectors, specifically $R_{i,\theta} q_i$ and $R_{j,\theta} k_j$, mirroring the operation of absolute position encoding. Note, the rotation matrix is applied outside the FlashAttention2 (Dao, 2024) module using Float32 precision. But, the inner product of the key and query occurs within the FlashAttention2 module and must be cast to BFloat16, as required by FlashAttention2. And, the selection of rotation angles satisfies $\theta_i = base^{-2i/d}$, the typical *base* value for current LLMs is 10,000.

### 2.2 BFloat16 Disrupts RoPE's Relative Properties, Especially with Long Context

The RoPE is theoretically designed to depend solely on the relative positional distance $m = j - i$ between tokens. A key implication is that adding a constant positional shift $\Delta$ to every position index should not alter the attention computation. Formally, it can be expressed as:

$$A_{(i+\Delta)(j+\Delta)} = (R_{i+\Delta,\theta} q_i)^\top (R_{j+\Delta,\theta} k_j) = q_i^\top R_{(j+\Delta)-(i+\Delta),\theta} k_j = q_i^\top R_{m,\theta} k_j = A_{ij}, \tag{3}$$

where $A_{ij}$ is the attention logit between positions $i$ and $j$, $R_{m,\theta}$ is the rotation matrix corresponding to the relative distance $m$ (derivation in Appendix A). However, a key question is, does this relative property hold in practice?

We empirically investigate the impact of positional shifts on attention computations. In order to clarify our analysis, we redefine the attention computation as $\text{ATTN}^{l,h}(\mathbf{X}, \Delta)$, representing the attention matrix produced by the $h$-th head in the $l$-th transformer layer when a positional shift $\Delta$ is applied. Specifically, if the original position indices are $[0, 1, \ldots, L-1]$, after applying the shift they become $[\Delta, \Delta+1, \ldots, \Delta+L-1]$.

We measure the difference in attention due to two different positional shifts $\Delta_1$ and $\Delta_2$ using the following metric:

$$D(\mathbf{X}, \Delta_1, \Delta_2) = \sum_{l,h} \sum_{j=1}^{L} \mathbf{n} \odot \left( \sum_{i=1}^{L} \left| \text{ATTN}_{i,j}^{l,h}(\mathbf{X}, \Delta_1) - \text{ATTN}_{i,j}^{l,h}(\mathbf{X}, \Delta_2) \right| \right), \tag{4}$$

This metric measures the cumulative difference in attention scores across all layers and heads. And the normalization vector $\mathbf{n} = \left[ \frac{1}{L}, \frac{1}{L-1}, \ldots, 1 \right]$ is applied element-wise (denoted by $\odot$) to account for the varying number of elements in lower triangular matrices due to causal masking.

**Experimental Setup.** In Figure 1 (left), we set $\Delta_2 = 16$ and vary $\Delta_1$ over the list $[0, 2, 4, 6, 8, 10, 12, 14, 15, 17, 18, 20, 22, 50, 100, 200, 500, 1000, 2000]$. The blue line represents results obtained using the pretrained parameters of `LLaMA-2-7B` with BFloat16 precision. The yellow line shows results using the same pretrained parameters, but converted to Float32 precision. The green line corresponds to results when using random initialized parameters (initialized with `kaiming_uniform_` in PyTorch by default) in BFloat16 precision. For each case, we compute the difference $D$, averaging over 50 text sequences, each with a length of $T = 4096$. Note, we shop the $\Delta_1 = 16$ to improve visualization. As expected, the difference is zero when both $\Delta_1$ and $\Delta_2$ are equal. More detailed visualizations and discussions are provided in the Appendix B.

**Results Discussion.** We observed that when using BFloat16 precision, the positional shift $\Delta$ affects the attention computations of the pretrained `LLaMA-2-7B`. In contrast, when all computations are performed with Float32 precision, this effect disappears (as shown by the blue vs. green lines). Additionally, when comparing pretrained parameters to randomly initialized ones, both under BFloat16 precision (blue vs. yellow lines), we found that pretraining amplifies the discrepancy. These findings suggest that under BFloat16 precision, the RoPE deviates (slightly) from its theoretically claimed relative positional encoding, and this property breaks down after the model is pretrained.

Furthermore, we delve into the details of the attention difference between $\Delta_1 = 0$ and $\Delta_2 = 16$ for each individual token (i.e., without summing over $t_{\text{row}}$ in our metric for *per-token* study). We visualize these per-token differences in Figure 1 (middle). The results clearly indicate that the first token contributes most significantly to the attention difference. When the first token is excluded, the remaining tokens retain the relative positional property.

In the previous experiments, we fixed the sequence length at $T = 4,096$. To investigate how sequence length affects the attention logit difference, we extended our study to sequence lengths of $[64, 128, 256, 512, 1024, 2048, 4096, 8192]$. In this trial, we measured the attention logits instead of the attention scores because the softmax operation over varying sequence lengths would render the results for the first token incomparable across different lengths. For clarity, we define the attention logit difference as:

$$D_{\text{logit}} = \frac{1}{T} \sum_{l,h} \sum_{i=1}^{T} \left| A_{i,j=1}^{l,h}(\Delta_1) - A_{i,j=1}^{l,h}(\Delta_2) \right|, \tag{5}$$

where $A_{ij}^{l,h}(\Delta)$ is the attention logit matrix with positional shift $\Delta$ for the $l$-th layer's $h$-th head. We keep $\Delta_1 = 0$ and $\Delta_2 = 16$ to measure the metric $D_{\text{logit}}$ averaged over 50 sequences with different context lengths. Figure 1 (right) summarizes these results. We observe that as the sequence length increases, the attention logit difference for the first token also increases. This indicates that longer sequences exacerbate the discrepancies in the first token.

> **Summary of the Section:** *Under BFloat16 precision, the relative property of RoPE is broken. The discrepancies are primarily sourced from the first token, which significantly contributes to the breakdown. Additionally, as the context length increases, the impact of these discrepancies becomes more pronounced.*

# 3 AnchorAttention

In the previous section, we demonstrated that BFloat16 precision compromises the relative positional encoding of RoPE, with discrepancies becoming more pronounced as sequence length increases. Nevertheless, BFloat16 remains desirable in practice due to its computational efficiency, which is especially crucial for long-context extensions. The most effective strategy in these scenarios involves continuing pre-training on long-context data (Fu et al., 2024; Gao et al., 2024). However, this approach is challenging because attention mechanisms incur a quadratic computational cost relative to sequence length. To mitigate the increasing errors with longer sequences while retaining the advantages of BFloat16, we propose **AnchorAttention**.

## 3.1 Training Long-Context Models with Small Window Sizes

Training a long-context model typically requires processing long sequences. However, this direct way exacerbates precision errors under BFloat16, as shown by Figure 1 (Right) with the number of processed tokens increasing, the discrepancy will exacerbate. Reducing the sequence length could mitigate the precision issues, but it seems contradictory to train a long-context model with short window sizes. To reconcile this, we revisit techniques from the literature and we found that the *intra-document attention* (Zhao et al., 2024b) that mask out cross-document attention (as shown in Figure 2 Left) can be used to train a long-context model with less sequence length compared with full attention. Besides, empirically, the intra-document attention has been successfully applied in open-source LLM, like `LLaMA-3` series. A recent work by Gao et al. (2024) also verifies the effectiveness of intra-document attention show that masking out attention across document boundaries improves both the short and long-context performance.

## 3.2 Does the Discrepancy in RoPE under BFloat16 Impact Long-context Performance?

Previously, we identified a deviation in the relative positional encoding of RoPE when utilizing BFloat16, as evidenced by differences in attention scores. This observation prompts the question: does this discrepancy significantly affect long-context performance? Especially, considering intra-document attention appears to mitigate discrepancies in attention scores, the impact of BFloat16 might be negligible on long-context capabilities, when the model is trained with intra-document attention. To investigate this, we compare two types of position indices. The first assigns position IDs continuously from the start to the end of the sequence (Figure 2 Left), following the common practice in intra-document attention Zhao et al. (2024b); Gao et al. (2024). The second resets the position index to 1 within each document (Figure 2 Middle). Theoretically, both positioning schemes should yield identical results. This comparison is used to assess whether the deviation in relative positional encoding further leads to measurable differences in long-context performance.

**Experimental Setup.** We conducted experiments by training a 128K model based on the `LLaMA-2-7B` architecture using the Slimpajama dataset with intra-document attention and evaluated them on the widely used long-context evaluation benchmark RULER. These experiments compared scenarios with and without resetting the position IDs, aiming to assess the impact of position ID assignment on model performance. We defer the training and evaluation protocols to Section 5.

**Results Discussion.** The results are presented in Figure 3. We observe that resetting position IDs consistently enhances the model's long-context performance. In contrast, the conventional approach, which not resetting position IDs (Figure 2 Left), results in inferior performance. This performance difference may arise from assigning different position IDs to the first token of each document, introducing inconsistencies that can potentially confuse the model. By resetting the position IDs, we maintain a consistent positional relationship (first position ID is on the first token). Note, this analysis is based on the

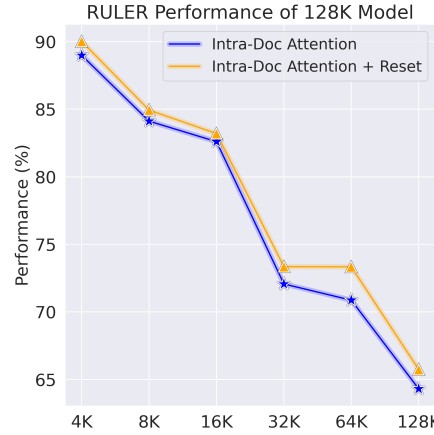

Figure 3: Resetting position IDs improves performance, contradicting theoretical predictions of RoPE.

hypothesis that RoPE functions in some way as an absolute positional encoding mechanism under BFloat16 precision. Further research is necessary to rigorously validate this assumption and fully understand its implications.

### 3.3 A Little Goes A Long Way: A Shared Anchor Token for Efficient Long-Context Training

Restarting position IDs enhances model performance but introduces a significant drawback: the model can only learn the full spectrum of rotational angles when processing sequences that reach or exceed the context length. This limitation hinders the model's ability to generalize to longer context length scenarios because, as we increase the context window size, collecting sufficient long sequences to fill the entire context window becomes impractical due to the scarcity of such lengthy data.

To address this challenge, we propose **AnchorAttention**, an attention mechanism that introduces a shared common anchor token visible to all documents within the long context window. Additionally, redundant attention connections across documents are ignored, allowing the model to focus on coherent information. An illustration of AnchorAttention is provided in Figure 2 (Right).

In AnchorAttention, the anchor token $\mathcal{A}$ serves as the common starting point for all documents. This design is motivated by our observation that the first token is responsible for deviations from relative positional encoding in RoPE; except for the first token, the subsequent tokens can preserve the desired relative positional encoding properties. By introducing a shared anchor token with a fixed position ID (the starting ID), the model is no longer confused about the relationship between the beginning of a document and the starting position ID. Specifically, we designate the beginning-of-sequence token, `<bos>`, as this common anchor. Since the `<bos>` token typically lacks explicit semantic meaning, any precision errors caused by BFloat16 concentrated on this token have minimal impact on overall model performance.

Furthermore, with the shared anchor, we eliminate the need to restart position IDs to ensure that the model correctly interprets the relationship between the beginning of sentences and the starting position ID. This allows us to use continuous position IDs from the start to the end of the window length (the non-resetting scheme, as shown in Figure 2, Right). Consequently, a full spectrum of rotational angles can be trained in each iteration. Compared to the intra-document attention approach, our method does not heavily depend on abundant long sequences that can fill the entire training window length to train the full rotational span. This reduces the dependency on collecting and upsampling long-sequence data.

> **Summary of the Section:**
> 1. *Deviations in RoPE's relative positional encoding observed in attention scores under BFloat16 precision also affect the model's ability to handle long sequences effectively.*
> 2. *Resetting position IDs within intra-attention mechanisms improves long-context performance by maintaining consistent positional relationships.*
> 3. *A Little Goes A Long Way: Introducing a shared anchor token provides a simple yet powerful solution for long-context training.*

## 4 Long-Context Extension Protocol

In this section, we focus on the optimized training strategies and robust evaluation metrics, detailing the selection of base models, the long-context training dataset, specific training configurations, and the benchmark setup.

### 4.1 Base Model, Dataset and Training Configuration

**Base Models.** In our experiments, we primarily use the `LlaMA-2-7B` model as the base model. To assess the effectiveness of our proposed methods across different pretrained models, we also evaluate `Llama-3-8B` (Dubey et al., 2024), `Qwen-1.5 (1.8B)` (Yang et al., 2024), and `Mistral-7B-v0.3` (Jiang et al.,

2023) in Section 5.3. These models represent a range of architectures and base model pretraining paradigms, allowing us to test the generality of our approach.

**Dataset.** We use the SlimPajama dataset (Soboleva et al., 2023) for long-context training, an open-source replication of the LLaMA pretraining data mixture (Touvron et al., 2023). The dataset includes diverse text sources, such as Common Crawl (CC), C4, GitHub, ArXiv, Books, Wikipedia (Wiki), and StackExchange. In addition to the original SlimPajama, we apply the data mixture method from Fu et al. (2024), which upsamples long sequences within each source while maintaining the overall distribution of the original dataset. This upsampling aims to better expose the model to long-context scenarios during training.

In our experiments, we sample 2 billion tokens from both the original and upsampled SlimPajama datasets. We refer to the datasets with sequence lengths of 64K and 128K tokens as `SlimPajama-64K` and `SlimPajama-128K`, respectively. The upsampled dataset is denoted as `UpSampledMix-128K`. Detailed statistics and discussions on data engineering are provided in Appendix C.

**Training Configuration.** Our training hyperparameters are primarily based on (Zhang, 2023). All models are trained on 8 NVIDIA A100 GPUs. We set the learning rate to $2 \times 10^{-5}$ and use the AdamW optimizer with weight decay of 0.1, $\beta_1 = 0.9$, and $\beta_2 = 0.95$. Each model is trained for 2000 steps, which corresponds to approximately 1 epoch over the 2 billion token dataset. The batch size is set to 8, equating to 0.5 million tokens per batch for 64K context and 1 million tokens for 128K context lengths. Table 1 summarizes the training configurations.

Table 1: The training Configuration.

| Long-context Continuous Training | | |
|---|---|---|
| Data | UpSampledMix / SlimPajama128K/ SlimPajama64K | |
| | UpSampledMix-128K: | 58% CC, 20% C4, 7% GitHub, 6% ArXiv, 5% Books, 4% Wiki, 2% StackExchange |
| | SlimPajama-128K: | 53% CC, 27% C4, 5% GitHub, 5% ArXiv, 4% Books, 3% Wiki, 3% StackExchange |
| | SlimPajama-64K: | 54% CC, 25% C4, 5% ArXiv, 5% GitHub, 4% Books, 3% Wiki, 3% StackExchange |
| Model | Initialization: | Llama-2-7B / Llama-3-8B / Qwen-1.5-1.8B / Mistral-7B-v0.3 |
| | RoPE: | 16K: $1 \times 10^6$, 64K: $5 \times 10^6$, 128K: $1 \times 10^7$ |
| | Attention: | Full attention/ Intra-doc attention / Intra-doc attention with Reset AnchorAttention / AnchorAttention with Tag |
| Optim. | AdamW (weight decay = 0.1, $\beta_1 = 0.9$, $\beta_2 = 0.95$) | |
| | LR: | $2e-5$       Steps:       2000 steps |
| | Batch size: | 8 (0.5M token for 64K, 1M tokens for 128K) |

## 4.2 Controllable Study with Meaningful Evaluations

**Measuring Long-Context Ability with Appropriate Metrics.** To conclusively evaluate the effectiveness of our proposed attention mechanism in enhancing the long-context capabilities of base models, it is important to use robust and suitable evaluation metrics.

Perplexity (PPL) is commonly used to evaluate long-context language models; however, recent studies question its reliability for assessing long-text understanding. Hu et al. (2024) show that PPL poorly correlates with a model's ability to comprehend long-range dependencies because it primarily measures local information capture. Additionally, Fang et al. (2024) provide empirical evidence that PPL overlooks key tokens crucial for understanding long-context inputs, leading to unreliable assessments. Moreover, Gao et al. (2024) find that while increasing the amount of long data improves PPL, exclusively using long data can significantly degrade downstream long-context performance. In line with these observations, we also found that while PPL remains unchanged after the initial training steps, performance on the RULER benchmark (Hsieh et al., 2024) continues to improve (in Figure 4), further indicating that PPL may not adequately reflect enhancements in long-context performance.

Existing benchmarks for long-context language models—such as HELMET (Yen et al., 2024), Long-Bench (Bai et al., 2023), LongBench-Cite (Zhang et al., 2024a), InfiniteBench (Zhang et al., 2024c), and NoCha (Karpinska et al., 2024)—heavily rely on instruction-following abilities, making them unsuitable for

evaluating long-context base models. Moreover, tasks in LongBench (Bai et al., 2023) can be adequately addressed with a context length of 16K tokens. While 16K tokens represent a *medium-length* context, this does not fully challenge models like ours that can handle 64K or even 128K tokens. Despite these limitations, we still evaluated our models on LongBench because it includes real-world long-context tasks, and its few-shot in-context learning (ICL) tasks are appropriate for assessing base models. Notably, the effectiveness of few-shot ICL in evaluating base models is also observed in concurrent work by Gao et al. (2024).

To more thoroughly assess our model's long-context capabilities, we employ the RULER benchmark (Hsieh et al., 2024), which is specifically designed for extensive context lengths. RULER evaluates abilities such as (1) locating specific data within vast content, (2) tracing relationships across broad contexts, (3) aggregating and quantifying dispersed information, and (4) distinguishing relevant details from extraneous information in complex queries. The benchmark includes tasks across various categories—Needle-in-a-Haystack (NIAH), Variable Tracing (VT), Common and Frequent Words (Aggregation), and Question Answering (QA)—each targeting different aspects of long-context processing that more accurately represent the capabilities of models like ours.

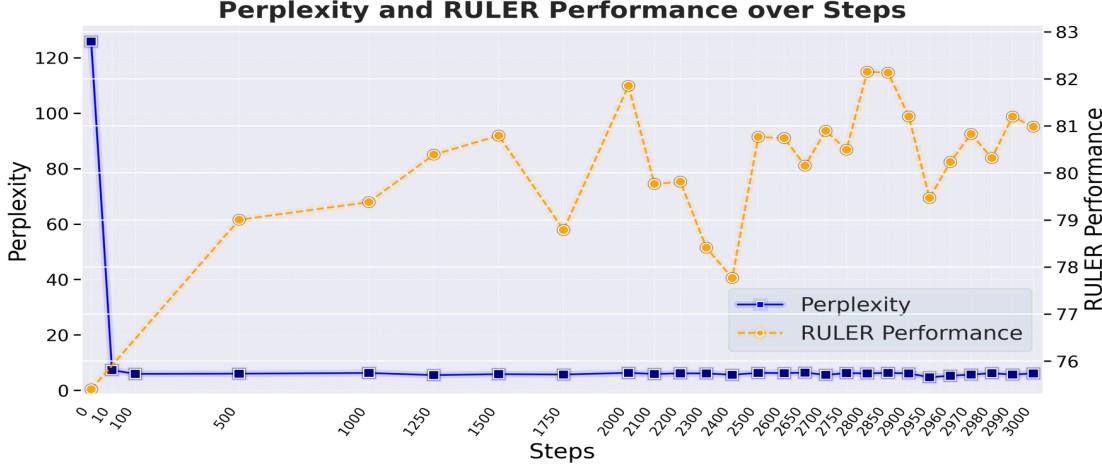

Figure 4: RULER performance varies during long-context training, we recommend reporting the averaged RULER performance rather than just the final training step. PPL remains unchanged after the first several steps, failing to reflect improvements in long-context ability.

However, unlike the official evaluation, which measures overall performance across all 13 tasks, we exclude two tasks (and NIAH-Multikey 3 and Common Word Extraction) for the `LLaMA-2-7B` model. This exclusion is based on the assumption that long-context training primarily extends a model's existing abilities from its original context window to longer sequences, and on the observation that `LLaMA-2-7B` struggled with these two tasks even within its original 4K context window. It is therefore unreasonable to expect the model to acquire new skills in extended contexts that it could not perform within its original window. (Detailed results for both `LLaMA-2-7B` and `LLaMA-2-7B-Chat` across all 13 tasks along with a comprehensive discussion are provided in Appendix D). Note that for other advanced models (`LLaMA-3-8B`, `Mistral-7B-v0.3`, `Qwen-1.5-1.8B`) in the cross-architecture evaluation section (Section 5.3), we use all 13 tasks.

Moreover, unlike previous studies that report RULER performance at a single checkpoint, we suggest reporting the average RULER performance across five checkpoints, saving a checkpoint every 10 steps over the last 50 training steps. Although retraining the model five times with different random seeds is another reasonable method for averaging performance, it would significantly increase the experimental cost. To demonstrate the necessity of averaging over multiple checkpoints, we evaluate the RULER performance of `LLaMA-2-7B` trained with standard full attention on the original SlimPajama with a 16K window size. As shown in Figure 4, the performance on the 16K RULER benchmark exhibits fluctuations during training. To prevent cherry-picking specific checkpoints, we suggests that future research report the averaged performance across several checkpoints.

**LLLM General Ability.** To ensure that continual pretraining preserves the models' core abilities within their initial context length, we use the MMLU (Hendrycks et al., 2021) and HellaSwag (Zellers et al., 2019) benchmarks. MMLU assesses knowledge and reasoning across 57 diverse subjects, including STEM, humanities, social sciences, philosophy, law, and medicine, providing a broad measure of general understanding. HellaSwag tests commonsense reasoning by having models choose the most plausible continuation of a scenario, challenging their contextual prediction skills. These benchmarks are standard choices for evaluating foundational abilities, and we observe minimal performance differences between the base and chat models, supporting their suitability for evaluating base model capabilities.

## 4.3 Optimizing Positional Embeddings for Long-Context Training

To find the optimal positional embedding for long-context training, we trained `LLaMA-2-7B` on the SlimPajama-64K dataset, chunked into sequences of 64K tokens. We evaluated three methods: RoPE (Su et al., 2021), NTK-aware (LocalLLaMA, 2023), and YaRN (Peng et al., 2023). Following (Men et al., 2024), we varied the base value in the embedding formula $\theta_i = \text{base}^{-2i/d}$ (Equation 6), corresponding to the `rope-theta` parameter in Transformers (Wolf et al., 2020). NTK-aware and YaRN depend on the relative scale $s = T_{\text{new}}/T_{\text{origin}}$, where $T_{\text{origin}}$ is the original context size (4K context length) and $T_{\text{new}}$ is the extended size (64K context length), so we set $s = 16$. Following (Fu et al., 2024), which suggests that 1B tokens suffice for learning long-context abilities, we use 1B tokens in each training to balance training time and the evaluation of multiple models.

Table 2 shows the performance from 4K to 64K tokens on the RULER benchmark. Vanilla RoPE with a suitable base value performs best within the training context length, outperforming NTK-aware and YaRN methods at 64K context length. To measure performance beyond the training context length (generalization), we evaluated the models on the 128K RULER benchmark (Table 4). YaRN embeddings generalize better to unseen longer contexts; while vanilla RoPE performs well within the trained length, its performance declines at longer contexts. Recognizing the importance of the RoPE base value, we further studied `rope-theta` selection by training models with a 32K context length to save computational resources. As shown in Table 3, performance peaks at a specific theta value, with marginal gains beyond that point and slight declines when increasing theta further, aligned with observations from (Liu et al., 2023b; Men et al., 2024). Based on these findings, we adapt base theta values with context lengths: 1M for 32K, 5M for 64K, 10M for 128K, and 50M for 256K. This approach ensures optimal performance across different context lengths.

Table 2: Long-context models trained with vanilla RoPE outperform those with NTK or YaRN when evaluated within the extended context window size. The Free* indicates models evaluated directly without training.

| | Base | 64K | 32K | 16K | 8K | 4K |
|---|---|---|---|---|---|---|
| Free* | RoPE 10K | - | - | - | - | 85.29 |
| | YaRN 10K | 0.18 | 1.09 | 6.02 | 18.38 | 29.06 |
| RoPE | 1M | 66.89 | 70.79 | 77.02 | 84.64 | **90.42** |
| | 5M | 68.81 | **72.68** | **81.03** | **85.79** | 89.28 |
| | 10M | **69.43** | 71.01 | 81.01 | 85.28 | 88.13 |
| YaRN | 1M | 64.22 | 65.69 | 73.28 | 79.98 | 85.02 |
| | 5M | 63.48 | 64.47 | 75.29 | 79.90 | 84.01 |
| | 10M | 58.91 | 62.03 | 70.68 | 75.62 | 83.04 |
| NTK | 10K | 47.32 | 53.05 | 60.62 | 72.42 | 81.61 |
| | 1M | 62.47 | 70.49 | 75.87 | 83.48 | 86.41 |
| | 5M | 61.45 | 68.52 | 75.90 | 78.47 | 85.17 |
| | 10M | 58.75 | 65.95 | 72.27 | 77.43 | 83.99 |

Table 3: An appropriate *base* is crucial for long-context training; using values that are too large or too small can degrade performance.

| Base | 32K | 16K | 8K | 4K | Avg. |
|---|---|---|---|---|---|
| 500 | 8.52 | 16.43 | 21.18 | 37.06 | 20.80 |
| 10K | 15.21 | 28.79 | 43.61 | 83.03 | 42.66 |
| 200K | 68.91 | 76.67 | 85.64 | 88.00 | 79.81 |
| 600K | 74.61 | **81.79** | 85.03 | **89.36** | **82.70** |
| 900K | 75.05 | 81.62 | 85.22 | 88.65 | 82.64 |
| 5M | **76.09** | 77.15 | **85.66** | 87.00 | 81.48 |
| 1B | 73.97 | 76.48 | 81.09 | 82.33 | 78.47 |

Table 4: Evaluating beyond the extended context window, YaRN show advantages.

| | Base | 128K | 64K |
|---|---|---|---|
| RoPE | 1M | 35.93 (-30.95) | 66.88 |
| | 5M | 53.98 (-14.83) | 68.81 |
| YaRN | 10K | 27.25 (-22.84) | 50.09 |
| | 1M | **57.07 (-7.15)** | 64.22 |
| | 5M | 53.30 (-11.18) | 64.48 |
| | 10M | 50.85 (-8.06) | 58.91 |
| NTK | 10K | 24.31 (-23.01) | 47.32 |
| | 1M | 35.53 (-26.94) | 62.47 |
| | 5M | 41.87 (-19.58) | 61.45 |

**Summary of the Section:**

1. *The RULER benchmark is effective for evaluating the long-context training of base models. But, it is advisable to average performance over last several checkpoints to mitigate variance.*

2. *(Vanilla) RoPE is effective, but you need to carefully select the base value.*

   - *Vanilla RoPE with a properly chosen base value excels within the training context length but may underperform beyond it.*
   - *YaRN offers better generalization to contexts longer than the training window.*
   - *A good base value for ALL (RoPE-based) positional embeddings is critical to achieve optimal performance; values that are too large or too small can degrade performance.*

## 5 Experimental Results

### 5.1 AnchorAttention Performance on RULER

We continued pretraining the `LLaMA-2-7B` model on three datasets—SlimPajama-64K, SlimPajama-128K, and UpSampledMix-128K—using various attention mechanisms, including Full Attention, Intra-Document Attention, and our proposed AnchorAttention. We then evaluated their performance on the RULER benchmark (Table 5). We observed that resetting positional IDs generally improved the performance of Intra-Document Attention, except in some cases where the differences were negligible (e.g., at 4K tokens for the SlimPajama-64K and UpSampledMix-128K datasets). Our proposed AnchorAttention consistently outperformed Full Attention and Intra-Document Attention across all datasets and sequence lengths, particularly excelling at longer contexts. On all three datasets, AnchorAttention achieved top scores at every length, consistently outperforming other baseline methods. We also observed that the UpSampledMix-128K dataset improves model performance when training with Full Attention and Intra-Document Attention mechanisms. However, when using our proposed AnchorAttention for long-context training, the performance gap between models trained on SlimPajama-128K and UpSampledMix-128K is significantly reduced. This suggests the potential for AnchorAttention to reduce dependency on carefully upsampled data, simplifying future data preparation processes.

### 5.2 What Works and Doesn't in AnchorAttention: Domain Tagging and Interleaved Chunks

We further investigate two data utilization strategies that may enhance the performance of AnchorAttention: domain tagging and interleaved chunks, as illustrated in Figure 5.

**Domain Tagging**: This strategy prepends each training text sequence with a domain identifier (e.g., "Wikipedia" or "CommonCrawl"), masking out the loss from these tags during training. This allows the model to recognize and potentially prioritize information from specific domains. Previous studies (Allen-Zhu & Li, 2024; Zhang et al., 2024b) suggest that domain tagging can optimize knowledge storage and aid in selective learning, retaining information while avoiding conflicts.

**Interleaved Chunks**: Documents are segmented into multiple chunks at random split points, shuffled, and recombined into new sequences, with the original order of chunks within each document preserved (e.g., the second chunk of a document always appears after the first chunk in the newly organized data sequence). Zhao et al. (2024a) employed this technique to generate synthetic long-context data, effectively training models to handle extended contexts.

In Table 5, our experimental results show that incorporating interleaved chunks (*AnchorAttention + Interleaved Chunks*) consistently degrades performance compared to the base AnchorAttention. To investigate whether this is due to an incompatibility between AnchorAttention and interleaved chunks, we conducted additional experiments using intra-document attention with interleaved chunks on the SlimPajama-64K and SlimPajama-128K datasets. This combination also resulted in worse performance, even falling below that of the baseline Full Attention method. Notably, previous work (Zhao et al., 2024a) demonstrated that interleaved chunks combined with Full Attention can improve performance. Therefore, we hypothesize that the

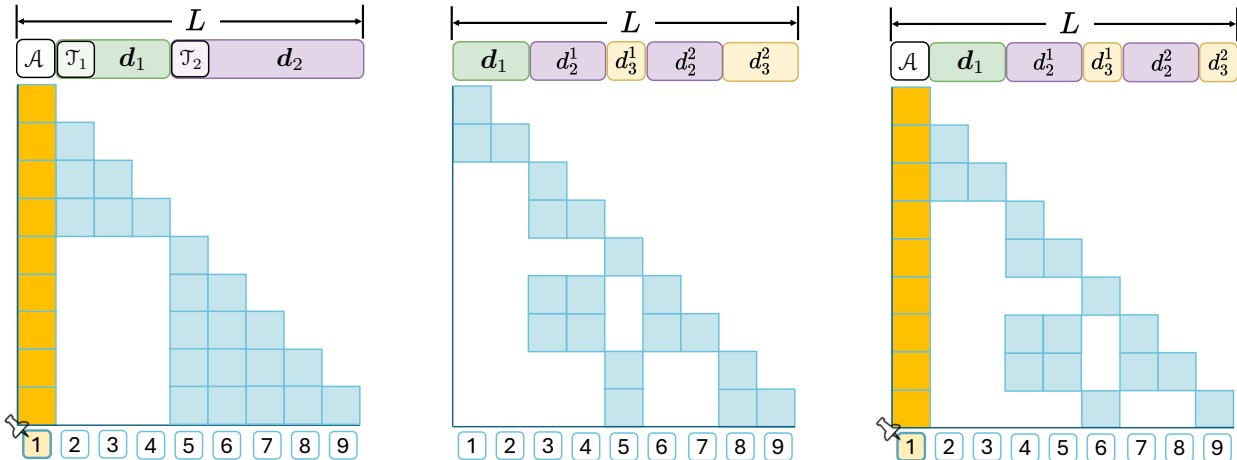

Figure 5: Illustrations of domain tagging and interleaved chunks. Left: AnchorAttention with domain tagging, where $\mathcal{T}_1$ denotes the domain of document $d_1$. Middle: Intra-document attention with interleaved chunks; documents are split into shuffled, interleaved chunks, preserving the original order within each document. Right: AnchorAttention with interleaved chunks.

cross-document attention masking is incompatible with interleaved chunks. Our proposed AnchorAttention generally improves performance over the baseline Full Attention but is less effective than using raw data, the coherence of documents preserved, with AnchorAttention. Regarding domain tagging, adding domain tags (*AnchorAttention + Tag*) does not consistently improve upon the base AnchorAttention. For example, on the SlimPajama-64K dataset, it achieves a slightly higher score at 64K tokens (73.88 vs. 73.25), but at 32K tokens, it performs slightly worse. Similar trends are observed on the SlimPajama-128K and UpSampledMix-128K datasets; *AnchorAttention + Tag* sometimes surpasses AnchorAttention at specific token lengths, but the base AnchorAttention generally performs better overall. Overall, those observations suggest that incorporating domain tagging can sometimes improve long-context performance, interleaved chunks consistently degrade performance when used with cross-document attention masking.

## 5.3 Cross-Model Evaluation of AnchorAttention for Long-Context Performance

To evaluate the generalizability of AnchorAttention across various model architectures, we assessed its long-context performance on multiple pretrained models. As shown in Table 6, our proposed *AnchorAttention* mechanism consistently outperforms the standard *Full Attention* across different models and context lengths, particularly with longer sequences. For all three models, `LLaMA-3-8B`, `Mistral-7B-v0.3`, and `Qwen-1.5-1.8B`, AnchorAttention yields higher scores across most sequence lengths. Note, the Qwen series does not utilize a `bos` token, we use the `eos` token as the shared anchor in our implementation. In the case of the `LLaMA-3-8B` model with a context length of 128K tokens, AnchorAttention achieves a score of 51.49 compared to 34.02 with Full Attention. Incorporating domain tags (*AnchorAttention + Tag*) sometimes further improves performance, as observed with the `Mistral-7B-v0.3` model at 128K tokens, where the score increases to 49.61. These results confirm that AnchorAttention effectively enhances long-context performance across different model architectures, demonstrating its robustness and broad applicability.

## 5.4 General Performance on Medium- and Short-Context Tasks

Enhancing long-context capabilities in language models often introduces a trade-off with performance on medium- and short-context tasks, as highlighted by previous research (Xiong et al., 2023). Our objective is not only to improve long-context understanding but also to ensure that our models retain strong performance on tasks with shorter contexts. To evaluate this balance, we tested our models on LongBench (a medium-context evaluation), as well as HellaSwag and MMLU (short-context benchmarks). Table 7 summarizes these results. The findings show that models trained with *AnchorAttention* better preserve general abilities

Table 5: Results on 64K and 128K Tokens Datasets. Highest scores across all methods are shown in **boldface**. Within the Intra-Doc Attention category, the higher scores are underlined. AnchorAttention and its variants, outperforming other methods, are highlighted with a background color .

| Attention Mechanism | 128K | 64K | 32K | 16K | 8K | 4K |
|---|---|---|---|---|---|---|
| *SlimPajama-64K* | | | | | | |
| Full Attention | \ | 66.40 | 71.78 | 77.63 | 83.86 | 89.84 |
| Intra-Doc Attention | \ | 69.97 | 74.70 | 79.15 | 83.50 | 89.62 |
| *+ Reset* | \ | 70.03 | 74.18 | 80.27 | 84.51 | 89.52 |
| *+ Interleaved Chunks* | \ | 60.59 | 66.52 | 71.70 | 79.70 | 84.71 |
| AnchorAttention | \ | 73.25 | **75.97** | **82.91** | **85.48** | **90.69** |
| *+ Tag* | \ | **73.88** | 74.21 | 82.46 | 85.13 | 89.93 |
| *+ Interleaved Chunks* | \ | 66.77 | 69.73 | 77.81 | 85.35 | 89.31 |
| *SlimPajama-128K* | | | | | | |
| Full Attention | 62.75 | 70.56 | 71.38 | 81.65 | 83.61 | 88.85 |
| Intra-Doc Attention | 64.31 | 70.87 | 72.07 | 82.60 | 84.11 | 88.98 |
| *+ Reset* | 65.75 | 73.34 | 73.30 | 82.82 | 84.43 | 90.01 |
| *+ Interleaved Chunks* | 53.74 | 61.08 | 65.51 | 75.25 | 80.59 | 82.71 |
| AnchorAttention | **66.15** | **77.69** | 74.28 | **83.67** | **86.41** | **90.60** |
| *+ Tag* | 65.46 | 74.67 | **75.77** | 83.07 | 84.07 | 89.09 |
| *UpSampledMix-128K* | | | | | | |
| Full Attention | 63.70 | 71.45 | 72.69 | 82.57 | 84.55 | 90.08 |
| Intra-Doc Attention | 63.96 | 74.52 | 76.53 | 82.46 | 86.61 | 90.35 |
| *+ Reset* | 64.10 | 74.55 | 77.73 | 82.82 | 87.16 | 89.98 |
| AnchorAttention | 65.24 | **76.11** | **79.51** | **86.54** | **87.43** | **90.44** |
| *+ Tag* | **66.85** | 73.52 | 77.18 | 81.62 | 84.90 | 89.01 |

Table 6: Attention Mechanism Performance Across Different Models and Token Sizes

| Attention Mechanism | 128K | 64K | 32K | 16K | 8K | 4K |
|---|---|---|---|---|---|---|
| LLaMA-3-8B | | | | | | |
| Full Attention | 34.02 | 61.80 | 72.09 | 79.99 | 82.43 | 83.68 |
| AnchorAttention | **51.49** | **70.99** | 83.06 | 86.90 | 88.09 | 88.72 |
| *+ Tag* | 49.67 | 70.37 | **84.14** | **87.13** | **88.36** | **88.97** |
| Mistral-7B-v0.3 | | | | | | |
| Full Attention | 45.64 | 49.05 | 54.49 | 64.06 | 69.99 | 72.80 |
| AnchorAttention | 47.46 | **61.26** | **68.53** | **73.47** | **76.06** | **78.94** |
| *+ Tag* | **49.61** | 56.80 | 64.13 | 69.47 | 74.65 | 77.34 |
| Qwen-1.5-1.8B | | | | | | |
| Full Attention | 33.56 | 41.77 | 47.01 | 56.15 | 61.33 | 67.26 |
| AnchorAttention | 34.32 | **44.31** | 48.63 | 56.90 | **62.62** | **68.61** |
| *+ Tag* | **35.84** | 43.91 | **50.70** | **57.39** | 61.96 | 67.41 |

from the pretraining stage compared to those using full attention or intra-document attention. Importantly, even when compared to the original LLaMA-2-7B, models with *AnchorAttention* remain competitive on short-context tasks. On the HellaSwag dataset, the *AnchorAttention* model trained on SlimPajama-64K achieves a score of 70.78, closely matching the original LLaMA-2-7B's score of 71.39. On the MMLU benchmark, the *AnchorAttention + Tag* variant attains a score of 42.85, nearing the baseline of 46.66. These results indicate that while significantly improving long-context capabilities, *AnchorAttention* effectively maintains performance on medium- and short-context tasks without substantial trade-offs. Additionally, incorporating domain tagging effectively preserves the general abilities of large language models.

Table 7: Results on LongBench ICL, HellaSwag, and MMLU datasets.

| Attention Mechanism | LongBench ICL | HellaSwag | MMLU |
|---|---|---|---|
| `LLaMA-2-7B` | 6.22 | 71.39 | 46.66 |
| *SlimPajama-64K* | | | |
| Full Attention | 62.51 | 68.50 | 33.93 |
| Intra-Doc Attention | 62.79 | **71.01** | 36.94 |
| *+ Reset* | 63.76 | 70.12 | 37.92 |
| AnchorAttention | 65.38 | 70.78 | 40.32 |
| *+ Tag* | **66.02** | 69.10 | **40.67** |
| *SlimPajama-128K* | | | |
| Full Attention | 50.72 | 69.46 | 37.93 |
| Intra-Doc Attention | 51.22 | 69.93 | 39.49 |
| *+ Reset* | 50.07 | 69.88 | 37.42 |
| AnchorAttention | 51.85 | **70.51** | 41.63 |
| *+ Tag* | **51.89** | 70.37 | **42.85** |
| *UpSampledMix-128K* | | | |
| Full Attention | 48.96 | 67.64 | 40.58 |
| Intra-Doc Attention | 49.51 | 70.86 | 41.27 |
| *+ Reset* | 50.18 | **70.97** | 40.79 |
| AnchorAttention | 50.17 | 70.11 | 41.15 |
| *+ Tag* | **50.70** | 68.97 | **42.03** |

## 5.5 Infrastructure and Engineering

Efficient implementation is crucial for training long-context language models. We introduce *AnchorContext*, a codebase that offers the *AnchorAttention* mechanism compatible with various models, including the LLaMA series, Mistral series, and Qwen2 series. To enhance compatibility with a wider range of frameworks, it provides two computational engine options: **FlexAttention** (which will be natively supported in PyTorch 2.5.0) and **FlashAttention** (currently the most commonly used).

To ensure the reliability of our experimental conclusions, we focused on comparing our method with existing techniques in terms of accuracy, speed, and ease of integration:

**Accuracy.** Distributed training can introduce numerical errors that affect model outputs, making it essential to assess whether a method maintains numerical consistency across different computational setups. We evaluated the numerical accuracy of our method by measuring the differences in logits (the outputs of the forward pass) when processing the same training data sequence with a context length of 32K on 8 A100 GPUs. Specifically, we compared the outputs of models using: *1. FlashAttention2* alone (the baseline, without distributed training), *2. Zigzag-Ring attention* (from the EasyContext implementation (Zhang, 2023)) in a distributed setting, *3. Our AnchorContext* based on sequence parallelism with DeepSpeed-Ulysses (Jacobs et al., 2023) in a distributed setting.

As shown in Table 8, the Zigzag-Ring attention exhibited a maximum logits difference of up to 0.75 compared to the baseline, indicating numerical discrepancies introduced by the distributed computation. In contrast, our *AnchorContext* implementation showed zero difference in logits when compared to using FlashAttention2 without any distributed training, highlighting its numerical accuracy.

Table 8: Our distributed computation achieves zero logits difference over 32K sequence length.

| | Zigzag-Ring (EasyContext) | Our Impl. (AnchorContext) |
|---|---|---|
| Full Attn | 0.75 | 0 |
| AnchorAttn | - | 0 |

**Speed and Efficiency.** Our method is not only accurate but also significantly faster. By integrating FlashAttention2 and optimizing the attention mechanism within our *AnchorContext* framework, we achieve higher GPU utilization and faster training times. Figure 6 illustrates the estimated number of days required to process 1 billion tokens at various context lengths. It demonstrates that our *AnchorAttention* substantially

reduces training time compared to Full Attention when using the same sequence parallelism and DeepSpeed-Ulysses configurations, showcasing the efficiency gains of our approach.

**Ease of Integration.** Designed with practicality in mind, our AnchorAttention seamlessly integrates into other training codebase that using FlashAttention (Dao, 2024) and Hugging Face transformers (Wolf et al., 2020). The flexibility of our *AnchorContext* approach allows for effortless adoption, enabling researchers to incorporate it to much substantial modifications to their codebase.

**Supporting Advanced Experiments.** Moreover, the combination of our method with FlexAttention mechanisms can support more advanced experiments. For example, it facilitates flexible attention masks needed by interleaved chunks, as illustrated in Figure 6. This flexibility empowers researchers to explore novel ideas and push the boundaries of long-context models.

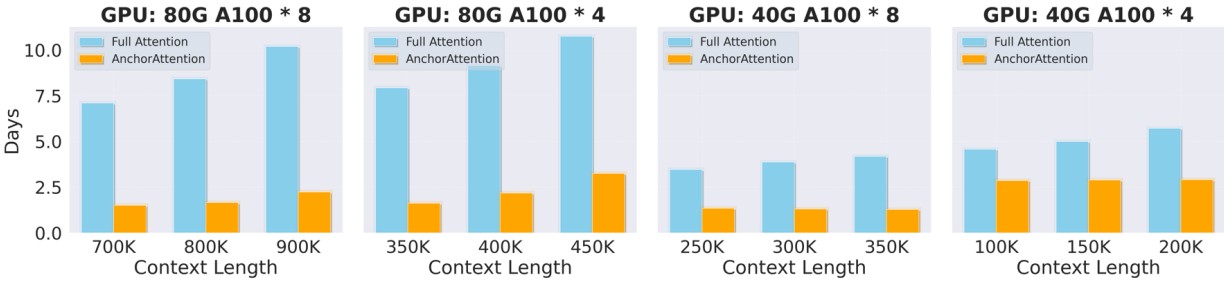

Figure 6: Estimated training time required to process 1 billion tokens at various context lengths using different attention mechanisms. Our *AnchorAttention* reduce more than 50% of time needed by Full Attention.

# 6 Related Work

**Positional Embedding.** Positional Embedding is essential for Transformers (Vaswani et al., 2017) to capture the sequential order of input tokens. Early methods used absolute or learned positional embeddings (Vaswani et al., 2017; Devlin, 2018), but they often struggled to generalize to longer sequences. Relative positional embeddings (Shaw et al., 2018; Ke et al., 2021) were introduced to handle variable-length sequences more effectively. Recently, Rotary Positional Embedding (RoPE) (Su et al., 2021) has become popular in LLMs (Touvron et al., 2023; Dubey et al., 2024; Jiang et al., 2023; Yang et al., 2024; Abdin et al., 2024) for its ability to extend inference context length with minimal fine-tuning (Liu et al., 2023b; Chen et al., 2023a; LocalLLaMA, 2023; Peng et al., 2023; Men et al., 2024; Fu et al., 2024). Building on RoPE, methods like Position Interpolation (Chen et al., 2023a), NTK Interpolation (LocalLLaMA, 2023), YaRN (Peng et al., 2023), Resonance RoPE(Wang et al., 2024b), and CLEX (Chen et al., 2024) have been proposed to enhance long-context capabilities. In a different direction, we investigate how the precision of BFloat16 could compromise the relative position properties of RoPE.

**Extending Language Model Context Lengths.** Although training-free methods have shown promise in extending context lengths Xiao et al. (2023); Han et al. (2023); Ruoss et al. (2023), recent studies indicate that dedicating additional training to handle long contexts yields significantly better results Fu et al. (2024); Xiong et al. (2023); Gao et al. (2024). The long-context training poses challenges: the quadratic computational complexity and performance degradation beyond the pre-training context length. To address these issues, various approaches have been proposed. Some methods extend context windows by training with modified RoPE (Chen et al., 2023a; LocalLLaMA, 2023; Peng et al., 2023; Men et al., 2024), adjusting the RoPE base to enable models to handle longer contexts with minimal training. Other strategies manipulate attention patterns to efficiently handle extended contexts (Chen et al., 2023b; Xiao et al., 2023; 2024; Bertsch et al., 2024; Jin et al., 2024; Ge et al., 2024; Yin et al., 2024), redesigning the attention mechanism to prioritize computational resources on the most relevant parts of the input, thus reducing overhead for longer sequences. Additional efforts reduce the computational complexity of attention through techniques like sparse attention (Lou et al., 2024; Ge et al., 2024), group query attention (Ainslie et al., 2023), with

optimize implementation (Dao, 2024). To further overcome hardware limitations, methods like Sequence Parallelism (Li et al., 2021) distribute sequences across multiple devices, while Ring Attention (Liu et al., 2023a) and DeepSpeed-Ulysses (Jacobs et al., 2023) enhance efficiency by optimizing communication strategies when processing segmented sequences. Our work adopts the continuous long-context training approach with efficient implementation, but specifically focuses on attention design to mitigate issues in RoPE caused by BFloat16 precision.

## 7 Conclusion

In this paper, we identified a critical issue where the combination of Rotary Position Embedding (RoPE) and BFloat16 precision breaks the relative positional encoding properties of RoPE, particularly in long-context scenarios. Our analysis revealed that the first token in the sequence contributes significantly to this breakdown, and the problem worsens as the training window size increases due to cumulative numerical errors. To address this, we proposed **AnchorAttention**, a novel attention mechanism that treats the first token as a shared anchor across all documents within the context window. By assigning the same position ID to this anchor token and ensuring it is visible to all documents while keeping tokens from different documents invisible to each other, AnchorAttention maintains consistency in positional encoding. This approach not only preserves the integrity of RoPE's relative positional properties but also reduces the number of tokens involved in attention computations, mitigating the accumulation of numerical errors. Our experiments demonstrated that AnchorAttention consistently outperforms full attention and standard intra-document attention methods on long-context benchmarks like RULER, across context lengths ranging from 8K to 128K tokens. Additionally, on real-world long-context benchmarks such as LongBench, AnchorAttention improved in-context learning performance while largely preserving the model's capabilities on general tasks like MMLU and HellaSwag. Importantly, AnchorAttention requires minimal modifications to existing training pipelines and reduces training time by more than 50% compared to standard attention training.

## 8 Future Work

In our empirical analysis, we measured the average attention difference over multiple sequences. We also examined the per-sample attention differences, with visualizations provided in the Appendix B. The differences were consistent across samples, and we observed that even a simple function could approximate the difference. Notably, the first token contributes the most to this discrepancy, leading us to hypothesize that the position ID of the first token acts as an absolute position. Future work is needed to more rigorously investigate the properties of the first position and its impact on positional encoding. Additionally, given the observed significance of the first token in an input sequence, we speculate a potential relationship between this phenomenon and the well-documented attention sinks and massive activation (Xiao et al., 2023; Han et al., 2023; Gu et al., 2024; Guo et al., 2024a; Sun et al., 2024; Guo et al., 2024b). Further research is required to explore this connection and to understand how it might affect attention mechanisms in long-context language models.

### Broader Impact Statement

The development of AnchorAttention addresses a critical challenge in natural language processing by enabling large language models (LLMs) to effectively handle long-context sequences. This advancement has the potential to significantly improve applications that require understanding and processing extensive textual data, such as financial document analysis, long-form content generation, and multi-document summarization. By enhancing the ability of LLMs to maintain coherence and context over long sequences, we contribute to the creation of more capable and contextually aware AI systems.

### Limitations

Although we try to ablate the major components of our proposed attention mechanism, due to resource limitations, we cannot exhaust all aspects, such as study its effectiveness for pretraining, optimization hy-

perparameters and additional data mixtures. We also limit ourselves to the 10B-scale model size regime with 2B tokens, which may limit the generalizability of our findings.

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

## A Rotary Positional Embedding (RoPE) and Relative Positional Encoding

Rotary Positional Embedding (RoPE) is a method employed to infuse positional information into transformer models by modifying the query ($q$) and key ($k$) vectors within the attention mechanism. RoPE enhances the model's ability to capture positional relationships through the following steps:

### A.1 Mechanism of RoPE

1. **Splitting into 2-Dimensional Chunks**: The query and key vectors are partitioned into 2-dimensional segments. This segmentation allows each pair of dimensions to undergo independent rotational transformations.

2. **Applying Rotational Transformations**: Each 2-dimensional chunk is rotated by an angle determined by a frequency parameter $\theta$. Specifically, for the $i$-th chunk, a rotation matrix $R_{i,\theta}$ is applied. This introduces positional information based on the token's position in the sequence.

The rotation matrix $R_{i,\theta}$ for the $i$-th chunk is defined as a block diagonal matrix composed of individual 2-dimensional rotation matrices, where the $d$ denotes the number of hidden dimension of each head, such as 128 for LLaMA-2-7B:

$$
R_{i,\theta} = \begin{bmatrix}
\cos(i\theta_0) & -\sin(i\theta_0) & 0 & 0 & \cdots & 0 & 0 \\
\sin(i\theta_0) & \cos(i\theta_0) & 0 & 0 & \cdots & 0 & 0 \\
0 & 0 & \cos(i\theta_1) & -\sin(i\theta_1) & \cdots & 0 & 0 \\
0 & 0 & \sin(i\theta_1) & \cos(i\theta_1) & \cdots & 0 & 0 \\
\vdots & \vdots & \vdots & \vdots & \ddots & \vdots & \vdots \\
0 & 0 & 0 & 0 & \cdots & \cos(i\theta_{d/2-1}) & -\sin(i\theta_{d/2-1}) \\
0 & 0 & 0 & 0 & \cdots & \sin(i\theta_{d/2-1}) & \cos(i\theta_{d/2-1})
\end{bmatrix}
\tag{6}
$$

### A.2 Incorporating RoPE into Attention Mechanism

In the attention mechanism, logits are computed based on the similarity between queries and keys. With RoPE, the rotated queries and keys are utilized in this computation as follows:

$$
A_{(i,j)} = \left(R_{i,\theta}\, q_i\right)^\top \left(R_{j,\theta}\, k_j\right) = q_i^\top R_{i,\theta}^\top R_{j,\theta} k_j
\tag{7}
$$

### A.3 Encoding Relative Positional Information

To explain how RoPE encodes relative positional information, consider the product of the rotation matrices $R_{i,\theta}^\top R_{j,\theta}$. Using trigonometric identities for sine and cosine of angle differences:

$$
\sin(\alpha - \beta) = \sin\alpha\cos\beta - \cos\alpha\sin\beta
$$
$$
\cos(\alpha - \beta) = \cos\alpha\cos\beta + \sin\alpha\sin\beta
$$

Focusing on a single 2-dimensional block, the product is computed as:

$$
\begin{aligned}
R_{i,\theta}^\top R_{j,\theta} &= \begin{bmatrix} \cos(i\theta_0) & \sin(i\theta_0) \\ -\sin(i\theta_0) & \cos(i\theta_0) \end{bmatrix} \begin{bmatrix} \cos(j\theta_0) & -\sin(j\theta_0) \\ \sin(j\theta_0) & \cos(j\theta_0) \end{bmatrix} \\
&= \begin{bmatrix} \cos\left((i-j)\theta_0\right) & \sin\left((i-j)\theta_0\right) \\ -\sin\left((i-j)\theta_0\right) & \cos\left((i-j)\theta_0\right) \end{bmatrix} \\
&= R_{m,\theta}^\top
\end{aligned}
$$

Here, $m = i - j$ represents the relative position between the query and key. This derivation demonstrates that the product $R_{i,\theta}^\top R_{j,\theta}$ depends solely on the relative position $(i - j)$, rather than the absolute positions of $i$ and $j$ individually.

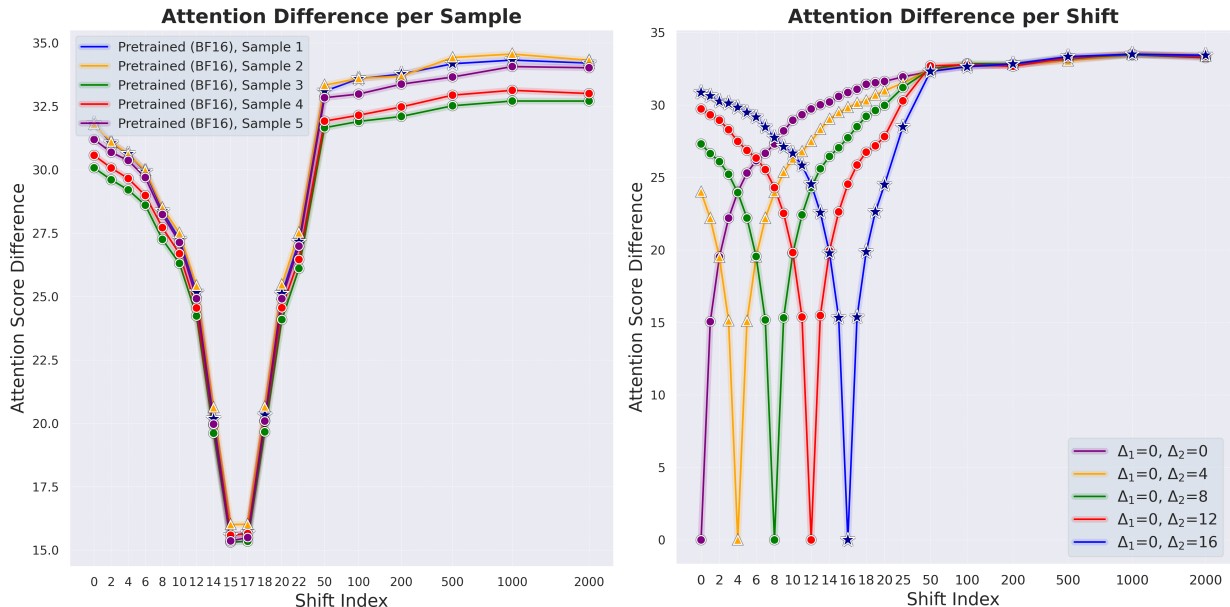

Figure 7: Detailed visualization of attention score differences under BFloat16 for individual samples.

## B   Detailed Analysis of Attention Discrepancies with RoPE under BFloat16

In Section 2, we primarily measured the attention differences averaged over multiple sequences. To provide a more detailed perspective, we present visualizations for individual samples. Figure 7 (left) shows the attention score differences for five samples, offering a detailed view compared to the averaged version represented by the blue line in Figure 1 (left).

As illustrated in Figure 7 (left), the attention score differences are highly consistent across different samples. The curves are nearly identical, exhibiting only slight biases for each sample. Given the starting differences ($\Delta_1 = 0$ and $\Delta_2 = 16$), we can accurately predict the attention differences for various $\Delta_1$ values.

Furthermore, while Section 2 focuses on fixing $\Delta_2 = 16$ and varying $\Delta_1$ to study attention differences, we extend our analysis by fixing $\Delta_1 = 0$ and varying $\Delta_2$. The results are summarized in Figure 7 (right). We observe that the attention difference curve shifts along the x-axis as $\Delta_2$ changes. These findings demonstrate that the attention score differences are highly predictable.

Building on our previous observation that the first token's positional encoding deviates significantly, we became curious about its role in establishing positional references for subsequent tokens. Empirical findings led us to hypothesize that *the position ID of the first token functions as an absolute position.* A more detailed discussion of this hypothesis will be explored in future work.

## C   Data Statistics

We present the mixture ratios and token contributions from each domain in our dataset. The *Mixture Ratio* is calculated by dividing the number of sequences from each domain by the total number of sequences. The *Token Ratio* is determined by dividing the token count of each domain by the total token count of the dataset.

For reference, the original SlimPajama token mixture ratios are as follows: the dataset consists of 82% web data (67% from CommonCrawl and 15% from C4), 4.5% code (GitHub), 4.5% Wikipedia, 4.5% books, 2.5% *ArXiv*, and 2.0% StackExchange. Since this dataset closely mirrors that used to pretrain the LLaMA models,

Table 9: Domain and Token Distributions

| | | C4 | Arxiv | Github | StackExchange | CommonCrawl | Wikipedia | Books |
|---|---|---|---|---|---|---|---|---|
| *– Up-sampled Data Mixture* | | | | | | | | |
| **128K** | Mixture Ratio | 52.34% | 1.01% | 3.68% | 4.56% | 33.40% | 4.79% | 0.21% |
| | Token Ratio | 19.53% | 5.86% | 6.61% | 1.64% | 58.14% | 3.51% | 4.69% |
| *– Original SlimPajama* | | | | | | | | |
| **128K** | Mixture Ratio | 55.32% | 0.30% | 3.65% | 5.06% | 31.01% | 4.59% | 0.06% |
| | Token Ratio | 26.50% | 4.64% | 5.05% | 3.18% | 53.42% | 3.34% | 3.88% |
| **64K** | Mixture Ratio | 55.05% | 0.40% | 3.66% | 4.97% | 31.23% | 4.58% | 0.10% |
| | Token Ratio | 25.43% | 5.22% | 5.05% | 2.95% | 54.24% | 3.24% | 3.86% |

there is less concern about distribution shift during continual pretraining; therefore, many recent works have utilized it.

Comparing the data we used for training, as shown in Table 9, we observe several key differences:

- **C4 Dataset:** Our up-sampled data mixture has a slightly lower Mixture Ratio for C4 (52.34% vs. 55.32% in SlimPajama) but a more pronounced decrease in Token Ratio (19.53% vs. 26.50%). This indicates that while the number of sequences from C4 is comparable, they are shorter on average in our dataset.

- **CommonCrawl:** We increased the Mixture Ratio for CommonCrawl (33.40% vs. 31.01%) and observed a higher Token Ratio (58.14% vs. 53.42%). This suggests that CommonCrawl sequences in our dataset are not only more numerous but also longer, contributing significantly to the total token count.

- **ArXiv and Books:** The representation of ArXiv and Books is enhanced in our up-sampled mixture. ArXiv's Mixture Ratio increased from 0.30% to 1.01%, and its Token Ratio from 4.64% to 5.86%. Similarly, Books saw an increase in Mixture Ratio from 0.06% to 0.21% and in Token Ratio from 3.88% to 4.69%. These increases aim to enrich the dataset with more scholarly and literary content.

- **StackExchange and Wikipedia:** The proportions of StackExchange and Wikipedia remain relatively consistent between the two datasets, ensuring stable representation of community-driven and encyclopedic knowledge.

- **GitHub (Code Data):** The Mixture Ratios for GitHub are similar (3.68% in our dataset vs. 3.65% in SlimPajama), but the Token Ratio is slightly higher in our dataset (6.61% vs. 5.05%), indicating longer code sequences that could benefit code understanding tasks.

Overall, our up-sampled data mixture places greater emphasis on longer sequences from CommonCrawl and increases the diversity of content by up-sampling underrepresented domains like ArXiv and Books. This rebalancing is designed to enhance the model's ability to generalize across various content types and improve its performance on tasks requiring knowledge from specific domains.

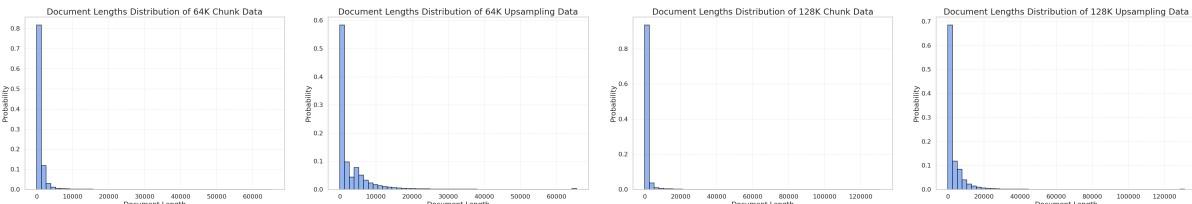

Figure 8: Training Data Sequence Length Distribution

| Model | NIAH Single 1 | NIAH Single 2 | NIAH Single 3 | NIAH Multikey 1 | NIAH Multikey 2 | NIAH Multikey 3 | NIAH Multivalue | NIAH Multiquery | VT | CWE | FWE | QA 1 | QA 2 |
|---|---|---|---|---|---|---|---|---|---|---|---|---|---|
| **Llama2 7B** | 100.0 | 100.0 | 99.8 | 97.2 | 87.8 | 44.0 | 99.1 | 99.35 | 59.0 | 24.46 | 91.73 | 61.2 | 43.0 |
| **+ Chat** | 95.2 | 100.0 | 99.8 | 93.2 | 90.0 | 70.2 | 95.8 | 98.7 | 88.4 | 34.26 | 85.93 | 64.8 | 39.4 |
| **+ Yarn 64K** | 73.0 | 24.4 | 8.0 | 18.0 | 5.8 | 0.8 | 5.9 | 6.35 | 54.2 | 18.16 | 57.8 | 38.6 | 27.6 |
| **+ Chat + Yarn 64K** | 67.4 | 48.8 | 32.4 | 30.2 | 16.4 | 4.8 | 48.0 | 34.75 | 54.16 | 43.48 | 82.07 | 41.2 | 25.0 |

Table 10: Results of different models across various tasks on $4,000$ context length.

## D `LLaMA-2-7B` RULER Performance

In Table 10, we use the base template for all models. Compared to the results reported in (Hsieh et al., 2024), we observe that Llama 2 7B's performance on the Common Word Extraction (CWE) task is unstable. When the context length is set to 4,000 (less than 4,096), a slightly different context example is provided (refer to the `generate_input_output` function in `scripts/data/synthetic/common_words_extraction.py` in RULER [2]). As shown in Table 11, when the context length is set to $4,096$, we can reproduce the reported results. However, the inclusion of a slightly different example disrupts the Llama 2 7B model's ability to accurately count words, highlighting the instability of the CWE task for evaluating Llama 2 7B. Consequently, we omit this task from the long-context evaluation.

Furthermore, based on our hypothesis that continued training of large language models primarily improves their ability to handle longer contexts rather than introducing new capabilities, we have excluded the NIAH Multikey 3 and CWE tasks from our evaluation. This decision is supported by the observation that even within the pretraining context length, Llama 2 7B is unable to effectively solve these tasks.

|  | **4,000** | **4,096** |
|---|---|---|
| `LLaMA-2-7B` | 24.46 | 76.8 |

Table 11: Performance of `LLaMA-2-7B` on Common Word Extraction (CWE) with different context lengths.

## E Longbench full result

---

[2]`https://github.com/NVIDIA/RULER/tree/0193bdd0abf6da542775f72b53352be8971d03f6`

| | Code Completion | ICL | Multi-Doc QA | Single-Doc QA | Summarization | Synthetic |
|---|---|---|---|---|---|---|
| *SlimPajama-64K* | | | | | | |
| Full Attention | 60.52 | 62.51 | 9.68 | 17.34 | 16.09 | 2.87 |
| Cross-Doc Attention | 62.95 | 62.79 | 9.51 | 16.82 | 16.73 | 2.94 |
| - *reset* | 62.76 | 63.76 | 9.30 | 16.40 | 14.61 | 3.74 |
| AnchorAttention | 62.04 | 65.38 | 9.72 | 18.60 | 17.56 | 4.24 |
| - *tag* | 63.53 | 66.02 | 9.51 | 18.28 | 15.30 | 5.24 |
| *SlimPajama-128K* | | | | | | |
| Full Attention | 54.17 | 50.72 | 6.36 | 16.43 | 13.30 | 2.04 |
| Cross-Doc Attention | 54.59 | 51.22 | 6.42 | 15.59 | 13.92 | 3.63 |
| - *reset* | 52.51 | 50.07 | 6.30 | 16.64 | 14.45 | 4.18 |
| AnchorAttention | 54.14 | 51.85 | 6.32 | 17.74 | 12.67 | 3.89 |
| - *tag* | 55.81 | 51.89 | 5.93 | 17.67 | 12.43 | 3.41 |
| *UpSampledMix-128K* | | | | | | |
| Full Attention | 53.13 | 48.96 | 6.12 | 14.66 | 12.77 | 4.13 |
| Cross-Doc Attention | 54.16 | 49.51 | 5.72 | 14.62 | 14.38 | 2.57 |
| - *reset* | 54.29 | 50.18 | 5.57 | 14.30 | 15.23 | 2.55 |
| AnchorAttention | 53.90 | 50.17 | 6.30 | 18.29 | 13.78 | 6.13 |
| - *tag* | 55.13 | 49.70 | 5.65 | 16.90 | 15.53 | 4.20 |

Table 12: Performance Metrics across Different Attention Mechanisms and Datasets.

