# OpenReview forum: "When Precision Meets Position: BFloat16 Breaks Down RoPE in Long-Context Training"
_TMLR — Accepted by TMLR_

### Review · Reviewer_QoHJ · 2024-12-19

**Summary Of Contributions:**

The paper introduces AnchorAttention, a novel attention mechanism designed to mitigate the limitations of RoPE when used with BFloat16 precision in long-context language models. The primary contributions are:
1. The authors identify that RoPE's relative positional encoding properties break down under BFloat16 precision, especially in long-context scenarios, due to cumulative numerical errors.
2. AnchorAttention introduces resetting position IDs for intra-document attention and a shared anchor token to alleviate numerical issues. This approach reduces the computational cost and improves performance on long context tasks.
3. Extensive experiments demonstrate that AnchorAttention reduces training time by over 50% while maintaining or improving performance on benchmarks like RULER and LongBench.

**Audience:**

Yes

**Claims And Evidence:**

Yes

**Requested Changes:**

See the weaknesses part.

**Strengths And Weaknesses:**

Strengths:
- The introduction of AnchorAttention is timely and addresses a critical issue in long-context model training with BFloat16 precision.
- The paper is easy to follow and well written.
- Authors provide extensive experimental results to demonstrate the effectiveness of AnchorAttention
- Authors release a useful package of AnchorAttention based on flashAttnetion, which can be useful for the community.

Weaknesses:
- (Minor) Though the piece of work provides very strong empirical results. More theoretical understanding can help readers to understand why the proposed method works.
- Question: What are the trade-offs of resetting the position IDs? Will it influence multi-document understanding (e.g. QA based on multiple documents)?

---

> ### Author Response · Authors · 2024-12-31
>
> We thank the reviewer for their insightful and encouraging feedback! We appreciate the acknowledgment that our work identifies "RoPE’s relative positional encoding properties break down under BFloat16 precision", especially in long-context scenarios, and introduces the motivated method AnchorAttention, which "reduces computational cost and improves performance on long-context tasks". We also value the reviewer’s comment on the "Authors release a useful package of AnchorAttention based on flashAttnetion, which can be useful for the community"! We address all questions below.
>
> ---
>
> ### **Weakness 1**
>
> > *"Though the piece of work provides very strong empirical results, more theoretical understanding can help readers to understand why the proposed method works."*
>
> **Answer:**
>
> Yes, we agree. Furthermore, a theoretical understanding (not only an empirical observation) of how the relative positional encoding properties break down under BFloat16 precision would deepen readers’ comprehension. We are currently working on another project that focuses on this phenomenon. In this work, we prefer to keep the focus on the long-context training. Thank you for your suggestion; we will try to incorporate a deeper analysis of the effectiveness of the proposed method, based on the theoretical understanding of the relative positional encoding breakdown in our future work.
>
> ---
>
> ### **Question 1**
>
> > *"What are the trade-offs of resetting the position IDs?"*
>
> **Answer:**
>
> By "resetting the position IDs" we believe you are referring to Intra-Doc Attention + Reset (the middle figure of Figure 2). The trade-off is that each new document starts at the same position ID, which provides consistency and avoids potential inconsistencies that might confuse the model. However, this means the maximum ID span is tied to the length of the longest document (because each document’s IDs are reset, so the maximum ID is effectively the maximum document length). Consequently, the learned spectrum of rotational angles is restricted to this length. This limitation prompted us to propose AnchorContext, which does not require ID resetting, as shown in the right of Figure 2.
>
> ---
>
> ### **Question 2**
>
> > *"Will it influence multi-document understanding (e.g., QA based on multiple documents)?"*
>
> **Answer:**
>
> Multi-document QA capabilities are typically unlocked during the Supervised Fine-Tuning stage, whereas AnchorAttention focuses on the continuous pre-training phase. Therefore, AnchorAttention is not directly related to multi-document understanding. However, if you wish to apply AnchorAttention in the SFT stage, we suggest treating multiple documents and QA pairs together as one "document". If the attention between multiple documents and QA pairs is masked, then the QA ability cannot be improved in that scenario.

---

> > ### Comment · Reviewer_QoHJ · 2025-01-11
> >
> > Thank you for addressing my questions. My concerns are resolved.

---

### Review · Reviewer_y1GF · 2024-12-19

**Summary Of Contributions:**

RoPE has become the standard positional encoding method for training LLMs with long context windows as it allows for out-of-distribution generalization of relative positions. However, the paper shows that when RoPE is implemented in combination with brain floating point (bfloat16), commonly used to reduce memory requirement by reducing precision, the relative property of RoPE is compromised especially for long-context scenarios.

The paper further shows empirically that the discrepancy in attention calculation is mainly due to the first token in a sequence. To mitigate the effect of the discrepancy, the paper introduces AnchorAttention, which builds on intra-doc attention by including a special Anchor Token shared by all documents. AnchorAttention shows significant improvement on the RULER benchmark over full attention and intra-doc attention.

**Audience:**

Yes

**Claims And Evidence:**

Yes

**Requested Changes:**

Summary of weaknesses to address:

- Since AnchorAttention is motivated by the observation that the first token of a sequence contributes the most to the deviation of RoPE’s relative properties, can you show more experiments that support this claim? Figure 1 (middle) only shows this for a shift of 0 and 16, does this same drastic deviation happen with different deltas?

- Can you show the performance of float32 on the same benchmark in section 3.2 and figure 3? If not, can you justify why the current experiments are sufficient?

Minor comments and questions:

- I think the x-axis of Figure 1 (left) could be more intuitive if it shows the difference in the two deltas instead of the value of delta 1. I also think omitting delta 1 = 16 adds to some confusion, and it would be good to show that when the two deltas are equal the difference goes to zero as a sanity check.

- Are all experiment results in Section 4.3 conducted with bfloat16?

**Strengths And Weaknesses:**

Strength:

- BFloat16 breaking the relative property of RoPE is an important observation that is highly relevant to other researchers.

- AnchorAttention is simple to implement and shows improvement over full attention and intra-document attention on the RULER benchmark across multiple LLM architectures.

- The codebase provided is easily adaptable by the research community.


Weakness:

- I would have liked more intuition behind the observations extracted from the experiments in Section 2. Why is the discrepancy so large at the first token? Why is there a small uptick in the difference in attention at the end of the sequence (figure 1 middle)?

- Section 3.2 investigates the effect of the RoPE + BFloat16 discrepancy on long-context tasks, but the authors did not show the performance on the same benchmark using float32, which I think would be important to understand the effect of BFloat16. Is this drop in performance as the context length increases the result of BFloat16 or something else?

- The connection between AnchorAttention and the limitation of BFloat16 can be further clarified. Is AnchorAttention’s motivation to reduce the impact of the discrepancy introduced by BFloat16 instead of addressing the discrepancy itself?

---

> ### Author Response · Authors · 2024-12-31
>
> We thank the reviewer for their detailed and insightful feedback! We appreciate that you found our observations regarding BFloat16 breaking RoPE’s relative property "an important observation that is highly relevant to other researchers" and that you noted our proposed AnchorAttention is "simple to implement and shows improvement". Below, we address your concerns point by point.
>
> ---
>
> ### **Weakness 1**
> > *I would have liked more intuition behind the observations extracted from the experiments in Section 2. Why is the discrepancy so large at the first token? Why is there a small uptick in the difference in attention at the end of the sequence (Figure 1 middle)?*
>
> **Answer:**
> This is a great question! Our current work primarily share our observation that RoPE’s relative property breaks down under BFloat16 and show how can we do long-context training better under this breakdown. We also share your curiosity about why and when this discrepancy arises, and we are conducting follow-up work to investigate its deeper causes. We would like to provide some (informal) insights:
>
> 1. **Amplification by High Attention Scores:**
>    The attention difference (as shown in Figure 1 (middle) in the paper) seems correlated with the "attention sink"[1,2,3,4] and "loss in the middle"[5,6] phenomena, where attention tends to be high at the first few tokens and at the last few tokens. These high attention scores amplify numerical differences under BFloat16.
>
> 2. **Emergence During Training:**
>    At random initialization (Figure 1 (left), green line), the discrepancy at the first token is small. However, throughout pretraining, the model utilizes the first token as a shortcut for positional reference. Consequently, RoPE behaves more like an absolute position embedding for that first token, making its positional encoding deviate more significantly.
>
> 3. **Absolute-Like Position for the First Token:**
>    The first token effectively anchors the rest of the sequence’s positional references. And we think that is why the Transformers without position encodings (NoPE) [7] can be success (in which they stress the first token is bos token).
>
> Our overall hypothesis is as follows:
>
> At the beginning (random initialization), the discrepancy is relatively small and negligible. During training, the "attention sink" and "loss in the middle" phenomena emerge, amplifying the differences. Subsequently, the model begins to use the first token as an anchor for the sequence’s positional references, further magnifying this effect.
>
> We plan to systematically study these phenomena in future work to provide a more rigorous theoretical understanding.
>
> **References**
>
> [1] Xiao, G., Tian, Y., Chen, B., Han, S., & Lewis, M. (2023). Efficient streaming language models with attention sinks. arXiv preprint arXiv:2309.17453.
>
> [2] Han, C., Wang, Q., Xiong, W., Chen, Y., Ji, H., & Wang, S. (2023). Lm-infinite: Simple on-the-fly length generalization for large language models. arXiv preprint arXiv:2308.16137.
>
> [3] Gu, X., Pang, T., Du, C., Liu, Q., Zhang, F., Du, C., ... & Lin, M. (2024). When Attention Sink Emerges in Language Models: An Empirical View. arXiv preprint arXiv:2410.10781.
>
> [4] Guo, T., Pai, D., Bai, Y., Jiao, J., Jordan, M. I., & Mei, S. (2024). Active-dormant attention heads: Mechanistically demystifying extreme-token phenomena in llms. arXiv preprint arXiv:2410.13835.
>
> [5] Zhang, Z., Chen, R., Liu, S., Yao, Z., Ruwase, O., Chen, B., ... & Wang, Z. (2024). Found in the middle: How language models use long contexts better via plug-and-play positional encoding. arXiv preprint arXiv:2403.04797.
>
> [6] Fu, Y. (2024, March). How do language models put attention weights over long context? Yao Fu’s Notion. https://yaofu.notion.site/How-Do-Language-Models-put-Attention-Weights-over-Long-Context-10250219d5ce42e8b465087c383a034e?pvs=4
>
> [7] Wang, J., Ji, T., Wu, Y., Yan, H., Gui, T., Zhang, Q., ... & Wang, X. (2024). Length Generalization of Causal Transformers without Position Encoding. arXiv preprint arXiv:2404.12224.

---

> > ### Author Response · Authors · 2024-12-31
> >
> > ### **Weakness 2**
> >
> > > *“Section 3.2 investigates the effect of the RoPE + BFloat16 discrepancy on long-context tasks, but the authors did not show the performance on the same benchmark using float32. Is this drop in performance as the context length increases the result of BFloat16 or something else?”*
> >
> > **Answer:**
> >
> > Let’s clarify the goal of Section 3.2. That section investigates whether the breakdown of RoPE’s relative positional property has a non-ignoreable impact on long-context training and performance. In Section 2, we show that RoPE’s relative property is broken (via attention differences), but we still don’t know whether this breakdown has a non-ignoreable effect on long-context training. Hence, in section 3.2, we want to take one more step to study whether the breakdown of RoPE’s relative positional property has a non-ignoreable impact on long-context performance.
> >
> >
> > The study logic of Section 3.2 essentially follows a proof by contradiction. If the breakdown of RoPE’s relative property had an ignoreable effect on long-context performance, resetting RoPE would not change performance (noting that here, we only question the role of relative positional property; we believe RoPE still provides positional information for attention, no matter whehter relative positional property is broken). However, we do observe a consistent performance improvement upon resetting RoPE, suggesting that the breakdown of its relative property indeed has a non-ignoreable impact. Moreover, this is aligned with our expectations: if RoPE’s relative property is compromised, RoPE behaves more like absolute positional embeddings, so resetting position IDs creates a more consistent mapping between the document and position IDs (as further discussed in Section 3.2).
> >
> >
> > You propose comparing RoPE + BFloat16 with RoPE + Float32. From the perspective of verifying whether RoPE’s relative property has a non-ignoreable impact on long-context performance, this approach could also yield a concrete conclusion. Our earlier observations indicate that RoPE’s relative property remains intact under Float32, so we could switch to Float32 (where the property is preserved) and see if there is an improvement—leading to the same conclusion. However, BFloat16 is required for FlashAttn, which provides essential speed and memory savings for long-context tasks. Without FlashAttn, RoPE + Float32 would cause a memory blowup; if we used a smaller model to avoid memory issues, its long-context performance would be too poor to draw meaningful conclusions.
> >
> > (Besides, in our way to answer the question, we study the effect of resetting position IDs, which actaully pave the way for delivering AnchorAttention.)
> >
> >
> >
> > ---
> >
> > ### **Weakness 3**
> >
> > > *“The connection between AnchorAttention and the limitation of BFloat16 can be further clarified. Is AnchorAttention’s motivation to reduce the impact of the discrepancy introduced by BFloat16 instead of addressing the discrepancy itself?”*
> >
> > **Answer:**
> >
> > Thank you for the suggestion! We will clarify the connection between AnchorAttention and the limitations of BFloat16 in our revision. Rather than trying to eliminate the precision error (since the LLM models has already been pretrained with BFloat16 and we have to work with it), we aim to find ways to improve long-context training despite the discrepancy introduced by BFloat16.
> >
> > Because BFloat16 is both memory- and computation-efficient—important for extending context windows—and because most LLMs are already pretrained under BFloat16, we believe it is more practical to accept its inherent precision limitations. Our strategy focuses on accommodating these constraints and developing methods like AnchorAttention that improve performance on long-context tasks within this widely-used framework.
> >
> > ---
> >
> > ### **Requested Changes 1**
> > > *“Since AnchorAttention is motivated by the observation that the first token of a sequence contributes the most to the deviation of RoPE’s relative properties, can you show more experiments that support this claim? Figure 1 (middle) only shows this for a shift of 0 and 16; does this same drastic deviation happen with different deltas?”*
> >
> > **Answer:**
> >
> > We agree that more experiments would strengthen the claim! In fact, we do have additional experiments in Appendix showing similar phenomena for different deltas, presented in Figure 7. We chose not to include these in the main text to keep the focus on our proposed long-context training method. Moreover, we believe a deeper theoretical analysis is needed to fully explain these phenomena, which we plan to address in a separate work.

---

> > > ### Author Response · Authors · 2024-12-31
> > >
> > > ### **Requested Changes 2**
> > > > *“Can you show the performance of float32 on the same benchmark in Section 3.2 and Figure 3? If not, can you justify why the current experiments are sufficient?”*
> > >
> > > **Answer:**
> > >
> > > I think this request is related to the previous point (Weakness 2). As for why our current experiments are sufficient, we refer the reviewer to our response under Weakness 2. There, we explain how both your suggestion and our current approach address the key question of Section 3.2: whether the breakdown of RoPE’s relative positional property has a non-negligible impact on long-context performance. However, testing float32 faces several practical challenges, as discussed above. Besides, our current approach (via resetting) serves as a key step in delivering our final method—AnchorAttention.
> > >
> > > ---
> > >
> > > ### **Questions 1**
> > > > *“I think the x-axis of Figure 1 (left) could be more intuitive if it shows the difference in the two deltas instead of the value of delta 1. I also think omitting delta 1 = 16 adds to some confusion, and it would be good to show that when the two deltas are equal the difference goes to zero as a sanity check.”*
> > >
> > > **Answer:**
> > >
> > > Thank you for this suggestion! We will consider revising the x-axis in Figure 1 (left) to make the differences more intuitive. As for showing both deltas being equal (where the difference goes to zero), we do provide that result in Figure 7.
> > >
> > > ---
> > >
> > > ### **Questions 2**
> > > > *“Are all experiment results in Section 4.3 conducted with bfloat16?”*
> > >
> > > **Answer:**
> > >
> > > Yes. All experiments in Section 4.3 were conducted under BFloat16.
> > >
> > > ---
> > > ---
> > >
> > > Once again, we appreciate your thoughtful feedback!

---

> > > > ### Comment · Reviewer_y1GF · 2025-01-10
> > > >
> > > > I thank the authors for their detailed response, my concerns have been addressed.

---

### Review · Reviewer_CR2M · 2024-12-21

**Summary Of Contributions:**

This paper starts with an interesting prospective of the impact of float16 to rotary position embedding. The thesis introduces AnchorAttention, a method that effectively addresses the issue where the first token in RoPE plays a dominant role in attention differences when using BF16. It includes a wealth of experiments that confirm the superiority of AnchorAttention in long-context modeling.

**Audience:**

Yes

**Broader Impact Concerns:**

No specific concerns

**Claims And Evidence:**

No

**Requested Changes:**

1.	Provide experiments showing that the superiority only arises in bfloat16 training/inference (e.g., by comparing fp32 training & inference). Also it might help to show that AnchorAttention works for both intra-document and full attention.
2.	I am curious why the first token has the largest Attention Difference? This looks not straight forward for me，It would be beneficial to include some fundamental theoretical analyses.Also, in Figure 1, there is a abrupt pitfall of Attention Difference at shift index of 15, why this happens? I would hope to see some theoretical analysis on this issue.

**Strengths And Weaknesses:**

# Strengths

1.The analysis of the impact of bf16 on RoPE at various positions is both novel and meaningful, yielding intriguing results.

2.The paper provides extensive experiments, including both long and short benchmarks, and the conclusions align well with the observed phenomenon that the difference in attention increases with context length.


# Weaknesses

My main concern is about the motivation of AnchorAttention. The authors assert that AnchorAttention enhances the model by avoiding the breakdown of RoPE in bfloat16, but the experiments provided are insufficient to substantiate this claim. I am quite intrigued by Section 2 and look forward to seeing more experiments that validate the impact of Attention differences. However, I became quite confused while reading Sections 3, 4, and 5, as there is no mention of bfloat16. Although I am convinced that AnchorAttention is indeed beneficial for long-context modeling, there are several problems: First, the Attention Difference does not necessarily correlate with differences in final performance. Secondly, it is doubtful whether the functionality of AnchorAttention is related to precision or any other factors (might be the impact of the first token itself?). The argument would be significantly more compelling with additional experiments demonstrating that the superiority of AnchorAttention is evident only in bf16 training and inference. Also, if the analyses of Attention Difference are accurate, then it should be feasible for AnchorAttention to also benefit full attention, too.

---

> ### Author Response · Authors · 2024-12-31
>
> We greatly appreciate your thoughtful and detailed review. We are pleased that you find our study on the impact of bfloat16 on RoPE "both novel and meaningful" and that our experiments demonstrate "the superiority of AnchorAttention in long-context modeling".  Some of the misunderstandings made us reflect on how to clarify our approach more effectively. Some of the misunderstandings you raised prompted us to clarify our approach further. We have revised the manuscript accordingly (revisions are highlighted in blue). Below, we address each of your points, beginning with a key clarification.
>
> ---
>
> ### **Clarification**
>
> You mentioned that "AnchorAttention enhances the model by avoiding the breakdown of RoPE in bfloat16". We would like to clarify that AnchorAttention does not explicitly prevent the breakdown of RoPE’s relative property under bfloat16. Instead, AnchorAttention is our proposed solution to the question: **How can we train long-context models more effectively when RoPE’s relative property is already compromised by BFloat16?**
>
> A seemingly straightforward idea after observing this precision issue would be to fix it (e.g., by using float32 or or an approximation).
> However, this is often impractical due to memory constraints and because most large-scale open-source models rely on FlashAttention, which typically uses bfloat16. In fact, existing pretrained models were trained with bfloat16 already.
>
> Because bfloat16 is both memory and computation-efficient, especially benefits when extending context windows; and because most LLMs have already been pretrained under bfloat16, we believe it is more practical to accept its inherent precision limitations and focus on strategies that accommodate and work around them.
>
> Under these conditions, we propose AnchorAttention as a practical solution for better long-context training in bfloat16.
>
>
> ---
>
> ### **Weakness 1**
>
> > *My main concern is about the motivation of AnchorAttention. The authors assert that AnchorAttention enhances the model by avoiding the breakdown of RoPE in bfloat16, but the experiments provided are insufficient to substantiate this claim.*
>
> **Answer:**
>
> The clarification section addresses a key part of the question about our motivation, and we would like to expand on it here.
>
> In Section 2 of our manuscript, we show that bfloat16 leads to an undesirable breakdown of RoPE’s relative positional property, evidenced by the "attention difference". We then ask whether this breakdown actually affects long-context performance. Our results in Section 3.2 suggest that it does (based on a proof-by-contradiction approach). Specifically, if the relative positional property had no effect, resetting position IDs should not influence long-context performance. However, we observe consistent performance improvements when resetting position IDs (Note, this result is aligned with our expectations. If the relative positional property is compromised, RoPE acts more like absolute positional embeddings. Resetting position IDs then makes a consistent relationship between the document and the position IDs. We provide discussion of this in Section 3.2).
> However, purely resetting position IDs introduces a drawback: the model can only learn the full spectrum of rotational angles when processing sequences that reach or exceed the context length. This limitation restricts the model's ability to generalize across varying sequence lengths, especially shorter ones. To address this, we propose AnchorAttention as a solution.
>
> For the questions, Can we avoid precision issues practically? Should we avoid them (only during the continuous training phase)? Does resolving precision issues during the continuous training stage enough (then we can still use the pretrained models)? We believe all of them are good questions, but they are beyond the scope of our current work.
>
> ---
>
> ### **Weakness 2**
>
> > *Attention Difference does not necessarily correlate with final performance.*
>
> **Answer:**
>
> Attention difference does not necessarily correlate with long-context performance, and our work does not aim to establish such a connection or claim that our method "cancels out" attention difference to improve performance. Rather, the observed attention difference under bfloat16 serves as evidence of the breakdown of RoPE’s relative positional property. This insight motivates us to explore resetting position IDs for documents, which can improve long-context performance under bfloat16, as demonstrated in Section 3.2 (Figure 3). Building on this, we discuss the limitations of simply resetting position IDs and introduce AnchorAttention as a more effective approach for enhancing long-context training.

---

> ### Author Response · Authors · 2024-12-31
>
> ### **Weakness 3**
> > *Is AnchorAttention’s functionality related to precision or other factors?*
>
> **Answer:**
>
> This question can be addressed from two perspectives: Is AnchorAttention’s functionality related to precision? And is it related to other factors?
>
> For the first question, AnchorAttention is not designed to fix precision issues directly (as discussed earlier). Rather, its design acknowledges that bfloat16 leads to a breakdown of RoPE’s relative-position property, and our aim is to work around that breakdown to improve long-context training.
>
> For the second question, AnchorAttention's functionality is influenced by two main factors: **masking out cross-document attention** and **isolating the first position (the bos token)**. Our experiments show that both factors are significant. Comparing full attention to AnchorAttention demonstrates the impact of cross-document masking (Note that current LLMs like Llama2, Llama3, Mistral, etc., use the BOS token by default to indicate the sequence beginning. Thus, comparing them can control the variable and isolate the effects of design choice). Comparing AnchorAttention to intra-document attention highlights the importance of isolating the first position.
>
> ---
>
> ### **Weakness 4**
> > *If the analyses of Attention Difference are accurate, AnchorAttention should also benefit full attention.*
>
> **Answer:**
>
> AnchorAttention is incompatible with full attention in its current form because it explicitly masks cross-document attention and isolates the first token, which alters the attention pattern from that of full attention (see Figure 2, right, in our manuscript). Hence, it is not simply a module that can be applied *on top of* full attention without modifying the latter.
>
> ---
>
> ### **Question 1**
> > *Provide experiments showing that the superiority only arises in bfloat16 training/inference (e.g., by comparing fp32 training & inference).*
>
> **Answer:**
>
> We think that this question stems from the reviewer’s impression that "AnchorAttention enhances the model by avoiding the breakdown of RoPE in bfloat16", as discussed in previous QA rounds. Here, we provide additional clarification.
>
> A direct comparison of float32 versus bfloat16 is beyond the scope of our current study. Our work specifically focuses on the widely adopted bfloat16 training paradigm and explores how to improve long-context performance under this paradigm, rather than advocating a return to float32.
>
> Given that most large-scale open-source models are already trained in bfloat16, these precision issues cannot be easily "fixed" post hoc without incurring significant computational and memory costs. As such, our objective is to propose a practical solution that accommodates the inherent limitations of bfloat16 and works effectively within this widely-used framework.
>
> ---
>
> ### **Question 2**
>
> > *It might help to show AnchorAttention works for both intra-document and full attention.*
>
> **Answer:**
>
> This question is related to the weakness 4. Because each attention pattern is distinct, we consider them separately rather than combining them. AnchorAttention is designed as a standalone pattern, not as an addition to either full attention or pure intra-document attention.
>
> ---
>
> ### **Question 3**
> > *I am curious why the first token has the largest Attention Difference*
>
> **Answer:**
>
> This is indeed an interesting phenomenon, and we are curious about it as well. Additionally, we have observed that the last few tokens can exhibit a growing difference. A comprehensive exploration of why bfloat16 leads to such differences, particularly at the first token, requires deeper investigation and deserves a separate study. We would like to provide some (informal) insights here. We suspect that "attention sink" effects and "loss-in-the-middle" phenomena may play a role.
>
> 1. **Amplification by High Attention Scores:**
>    The attention difference (as shown in Figure 1 (middle) in the paper) seems correlated with the "attention sink"[1,2,3,4] and "loss in the middle"[5,6] phenomena, where attention tends to be high at the first few tokens and at the last few tokens. These high attention scores amplify numerical differences under BFloat16.
>
> 2. **Emergence During Training:**
>    At random initialization (Figure 1 (left), green line), the discrepancy at the first token is small. However, throughout pretraining, the model utilizes the first token as a shortcut for positional reference. Consequently, RoPE behaves more like an absolute position embedding for that first token, making its positional encoding deviate more significantly.
>
> 3. **Absolute-Like Position for the First Token:**
>    The first token effectively anchors the rest of the sequence’s positional references. And we think that is why the Transformers without position encodings (NoPE) [7] can be success (in which they stress the first token is bos token).
>
> We plan to systematically study these phenomena in future work to provide a theoretical understanding.

---

> > ### Author Response · Authors · 2024-12-31
> >
> > ### **Question 4**
> > > *Why is there an abrupt pitfall of Attention Difference at shift index 15?*
> >
> > **Answer:**
> >
> > As described on page 4 (Experimental Setup), we remove the index 16 for better visualization. When $\Delta_1 = \Delta_2 = 16$, the attention difference is effectively zero (because we fix the random seeds). For completeness, we provide the corresponding results in Figure 7 (right).
> >
> >
> > **References**
> >
> > [1] Xiao, G., Tian, Y., Chen, B., Han, S., & Lewis, M. (2023). Efficient streaming language models with attention sinks. arXiv preprint arXiv:2309.17453.
> >
> > [2] Han, C., Wang, Q., Xiong, W., Chen, Y., Ji, H., & Wang, S. (2023). Lm-infinite: Simple on-the-fly length generalization for large language models. arXiv preprint arXiv:2308.16137.
> >
> > [3] Gu, X., Pang, T., Du, C., Liu, Q., Zhang, F., Du, C., ... & Lin, M. (2024). When Attention Sink Emerges in Language Models: An Empirical View. arXiv preprint arXiv:2410.10781.
> >
> > [4] Guo, T., Pai, D., Bai, Y., Jiao, J., Jordan, M. I., & Mei, S. (2024). Active-dormant attention heads: Mechanistically demystifying extreme-token phenomena in llms. arXiv preprint arXiv:2410.13835.
> >
> > [5] Zhang, Z., Chen, R., Liu, S., Yao, Z., Ruwase, O., Chen, B., ... & Wang, Z. (2024). Found in the middle: How language models use long contexts better via plug-and-play positional encoding. arXiv preprint arXiv:2403.04797.
> >
> > [6] Fu, Y. (2024, March). How do language models put attention weights over long context? Yao Fu’s Notion. https://yaofu.notion.site/How-Do-Language-Models-put-Attention-Weights-over-Long-Context-10250219d5ce42e8b465087c383a034e?pvs=4
> >
> > [7] Wang, J., Ji, T., Wu, Y., Yan, H., Gui, T., Zhang, Q., ... & Wang, X. (2024). Length Generalization of Causal Transformers without Position Encoding. arXiv preprint arXiv:2404.12224.

---

> > > ### Comment · Reviewer_CR2M · 2024-12-31
> > > **Thanks for clarification**
> > >
> > > The response has resolved most of my questions. I understand that it is challenging to perform a practical comparison with float32. (Although I still believe it might be beneficial to conduct some light-weight experiments by directly comparing bfloat16 AnchorAttention, Standard Attention with float32 AnchorAttention and Standard Attention, e.g., by comparing perplexities. This would not influence my decision.)
> > >
> > > However, another question has occurred to me: What if we were to directly start from position=2 in intra-document attention?

---

> ### Author Response · Authors · 2024-12-31
>
> Thank you for your quick follow-up! We’re glad to see that our previous response resolved most of your questions.
>
> I’d like to discuss based on your additional points and experimental suggestions in more detail, and address your questions.
>
> ---
>
> Returning to our observation that the attention difference and relative positional property break down under certain conditions, we considered two potential approaches:
>
> 1. **Fix it** by introducing an approximate method.
> 2. **Accept it** and propose a method for more effective long-context training.
>
> As we mentioned previously, we have been taking the second direction because many current and future LLMs are pre-trained in BFloat16. Focusing on this second direction, in our view, can benefit a broader community of researchers and practitioners.
>
> Note, the first direction still holds promise. Please see Figure 7 in the Appendix: the left-hand plot shows the attention difference across several samples, while the right-hand plot shows delta taking on different values. We found that these differences might be predictable, which suggests a potential workaround: instead of assigning positional IDs as \(0, 1, 2, 3, \ldots\), we could use values like \(0.2, 1.31, 3.19, \ldots\) by carefully precomputing the rotation angle. This would be an approximation approach.
>
> If we follow this approximation strategy, your suggestion to compare: BFloat16 (our method) vs. Standard Attention, and Float32 (our method) vs. Standard Attention would be absolutely necessary. We are currently pursuing this as a separate line of research and would like to incorporate your suggestions in its experimental section!
>
>
> ---
>
> **Question:** What if we were to directly start from position=2 in intra-document attention?
>
> **Answer:** If we simply start intra-document attention at position ID=2, we believe the performance would be suboptimal. This approach still introduces inconsistencies between the position IDs and each document (as discussed in the manuscript). Moreover, it poses a risk because, during pretraining, the model sees position IDs starting from 0 coupling with each input sequence. Suddenly changing this pattern in the long-context training stage could disrupt the learned relationships.

---

### Decision · Action_Editor_e6in · 2025-01-27

**Recommendation:** Accept with minor revision

**Comment:**

The paper presents a valuable contribution to the field, and the requested revisions will further strengthen its impact. The identified weaknesses are addressable and do not detract from the overall significance of the work. The suggested certifications will further enhance the paper's credibility and contribution to the research community.

The following points can be taken into consideration for the minor revision:

1. While the empirical results are strong, the paper lacks a deep theoretical understanding of why the RoPE breakdown occurs under BFloat16 and how exactly AnchorAttention mitigates this issue. Reviewers CR2M and QoHJ specifically request more theoretical analysis.

2. Reviewer CR2M raised concerns about the motivation of AnchorAttention, suggesting a need for clearer articulation of how it relates to the BFloat16 precision issue. The authors' response clarifies this, but the revised manuscript should reflect this more explicitly.

3. Reviewer CR2M's question about directly starting from position 2 in intra-document attention warrants a brief discussion in the paper, even if it's just to explain why it's suboptimal.

4.  Incorporate other minor clarifications and suggestions from the reviewers as needed.

**Audience:**

Given the paper's focus on a critical aspect of LLM training, its novel findings, practical implications, and the release of a usable implementation, it is highly likely that a substantial portion of TMLR's audience would be interested in this work. The reviewers' positive feedback further reinforces this assessment.

**Claims And Evidence:**

The experimental results are thorough and support the effectiveness. While a deeper theoretical analysis would strengthen the paper further, the empirical evidence is sufficient to establish the validity of the claims. The reviewers' feedback, despite some constructive criticism, generally supports the paper's findings and contributions.

Reviewers CR2M and QoHJ point out the need for a deeper theoretical understanding of why the breakdown occurs and how AnchorAttention works. The authors acknowledge this and state they are working on a separate project to address the theoretical aspects.

Reviewer y1GF requests a direct comparison with Float32. The authors explain the practical challenges (memory blowup with Float32 due to lack of FlashAttn support) and argue that their focus is on improving the widely used BFloat16 paradigm. They also clarify that the logic of their experiments in Section 3.2 is sufficient to demonstrate the impact of RoPE's breakdown under BFloat16.

---

> ### Author Response · Authors · 2025-02-08
>
> Dear AE,
>
>
>
> Thank you for your thoughtful feedback. We also appreciate the reviewers' valuable suggestions. We have made the clarifications and updates recommended in our revised manuscript.
>
> Thanks again for your guidance throughout the review process!
>
>
>
> --
>
> Sincerely,
>
> The Authors
>
> Submission 3712